# PKHD1L1 is a coat protein of hair-cell stereocilia and is required for normal hearing

Xudong Wu [1,2,6], Maryna V. Ivanchenko [1,6], Hoor Al Jandal[1,3], Marcelo Cicconet [4], Artur A. Indzhykulian [1,5] & David P. Corey [1]

The bundle of stereocilia on inner ear hair cells responds to subnanometer deflections produced by sound or head movement. Stereocilia are interconnected by a variety of links and also carry an electron-dense surface coat. The coat may contribute to stereocilia adhesion or protect from stereocilia fusion, but its molecular identity remains unknown. From a database of hair-cell-enriched translated proteins, we identify Polycystic Kidney and Hepatic Disease 1-Like 1 (PKHD1L1), a large, mostly extracellular protein of 4249 amino acids with a single transmembrane domain. Using serial immunogold scanning electron microscopy, we show that PKHD1L1 is expressed at the tips of stereocilia, especially in the high-frequency regions of the cochlea. PKHD1L1-deficient mice lack the surface coat at the upper but not lower regions of stereocilia, and they develop progressive hearing loss. We conclude that PKHD1L1 is a component of the surface coat and is required for normal hearing in mice.

[1] Department of Neurobiology, Harvard Medical School, 220 Longwood Avenue, Boston, MA 02115, USA. [2] Decibel Therapeutics, 1325 Boylston Street, Boston, MA 02215, USA. [3] Northeastern University, 360 Huntington Avenue, Boston, MA 02115, USA. [4] Image and Data Analysis Core, Harvard Medical School, 43 Shattuck St, Boston, MA 02115, USA. [5] Department of Otolaryngology, Harvard Medical School and Massachusetts Eye and Ear, 243 Charles St, Boston, MA 02114, USA. [6]These authors contributed equally: Xudong Wu, Maryna V. Ivanchenko. Correspondence and requests for materials should be addressed to A.A.I. (email: inartur@hms.harvard.edu) or to D.P.C. (email: dcorey@hms.harvard.edu)

Mammalian hair cells are post-mitotic, terminally differentiated sensory cells of the inner ear. They carry a bundle of actin-based stereocilia on their apical surfaces, which is deflected on a nanometer scale by either sound or head movements. At the tips of hair-cell stereocilia is an extraordinarily sensitive mechanotransduction apparatus, which relies on shearing movement between the tips of adjacent stereocilia. The bundle is directionally sensitive as a consequence of the unique arrangement of stereocilia within the bundle, which form rows of increasing height.

A hair cells lives for the life of the organism and the shape of its bundle is retained for that time, although the proteins within it turn over[1], suggesting an exquisite regulation of actin filament length. Mutations in over 30 deafness genes are reported to alter the bundle morphology when mutated (reviewed in ref. [2,3]). A high level of molecular interaction between the various stereocilia proteins is apparently required to develop and maintain the bundle through an organism's life. Several types of stereociliary surface specializations[4] have a critical role in shaping the bundle throughout development, and keeping the proper stereocilia arrangement thereafter.

In particular, stereocilia are cross-linked by a number of transient or permanent links, identified with electron microscopy, which ensure bundle integrity (Fig. 1)[4]. These include tip links, ankle links, transient lateral links, horizontal top connectors, kinocilial links, tectorial membrane (TM) attachment crowns, and the stereociliary coat. This array of links is progressively refined, as many transient surface structures are replaced with mature ones. Molecular components of many links have been reported (PCDH15, CDH23, USH2A, VLGR1, PTPRQ, STRC); all of these proteins are encoded by genes associated with hereditary deafness, and disruption of these genes in mice is reported to disrupt the bundle morphology (reviewed in ref. [2,3,5–7]).

Although some types of links are present on the bundle very briefly (10–20 days), these transient structures have a key developmental role in bundle maturation. For instance, stereocilia bundles are splayed in the absence of the transient link components, PCDH15 and CDH23[8]. Similarly, disruption of the STRC protein in mouse leads to a mildly disarrayed hair bundle[9,10]. Tip links convey force to transduction channels[11,12]. In frogs, horizontal top connectors help maintain tight bundle cohesion[13], although mammalian cochlear bundles are less well connected[14]. The TM attachment crowns facilitate the attachment of the tallest row of outer hair-cell (OHC) stereocilia to the TM.

Stereocilia links are differentially sensitive to exposure to the $Ca^{2+}$ chelator BAPTA and the protease subtilisin[4,12], and

molecular identities have been assigned for some components of link complexes. Tip links contain CDH23 and PCDH15[15–20], as do transient lateral and kinocilial links[21]. Ankle links contain USH2A and ADGRV1[22,23], and PTPRQ likely contributes to the shaft connectors[24]. STRC is thought to participate in the TM attachment crown and perhaps horizontal top connectors[9,10]. A short isoform of NPTN, Np55, is needed for the stability of TM attachment to OHC stereocilia[25].

Other surface specializations unique to stereocilia include the stereocilia coat, which is transiently present at the surface of stereocilia. Little is known about its function or molecular identity. It is unknown whether this coat is formed by a variety of proteins, or multiple copies of the same one. The coat could facilitate bundle cohesion by contributing to transient links[4], or prevent stereocilia fusion by creating a repulsive coat on their surface[26]. Alternatively, it could just be a collection of proteins transiently located at the surface of stereocilia, waiting to form any of the stereociliary surface specializations: transient or permanent links, or TM attachments.

Uncovering the molecular identity of proteins forming stereocilia surface specializations is an important step in understanding the basic mechanisms by which the stereocilia bundles form, move upon stimulation, retain, and maintain their morphology, carry the bundle through the organism's life, and repair it after constant stimulation and overstimulation.

In this study, we report a novel stereociliary protein, Polycystic Kidney and Hepatic Disease 1-Like 1 (PKHD1L1), which we identified using our expression profiling data, filtered by size, cell-type expression, and predicted protein domain structure. We present results showing that PKHD1L1 localizes to stereocilia bundles, and participates in forming of the stereocilia surface coat, causing hearing loss when absent from the hair cells. We show that mice lacking PKHD1L1 have no stereocilia coat at their tips, suggesting this protein is essential for formation of the stereocilia coat at their tips. Our study is the first, to our knowledge, to explore the molecular identity of stereocilia coat and to identify one of the proteins contributing to it.

## Results

**Identification of PKHD1L1 in hair cells**. To identify proteins that form stereocilia surface specializations, we generated a translational database using the Ribotag mouse[27] crossed with the *Gfi1-Cre* mouse[28], in which epitope-tagged ribosomes can be isolated along with associated mRNAs. The database allowed us

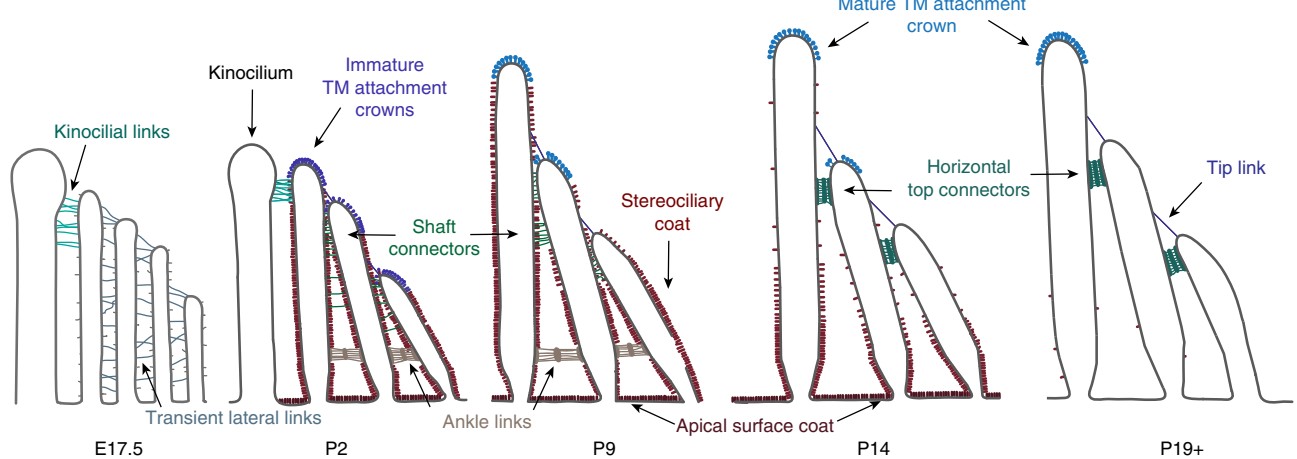

**Fig. 1** Links between stereocilia. Schematic illustration of the different types of links between the stereocilia in OHCs at different stages during development. Reproduced with permission from ref. [4]. 

to identify mRNAs that are actively translated, and preferentially translated in hair cells compared with non-hair cells. Details of the mRNA isolation and database will be presented separately. Based on other known stereocilia link proteins, we reasoned that novel link proteins would be large (>1500 aa), would be preferentially expressed in hair cells, and would have one or more transmembrane domains with most of the protein positioned extracellularly.

PKHD1L1 (Fig. 2a) was identified as having these characteristics as well as being highly enriched in hair cells. The database showed much higher levels of translated *Pkhd1l1* mRNA in hair cells at postnatal ages P0, P2, and P4, compared with translated *Pkhd1l1* mRNA in spiral ganglion cells or the total *Pkhd1l1* mRNA of the homogenized inner ear (Fig. 2b). Similarly, our previous database of all mRNAs isolated from hair cells[29] showed much higher transcription of *Pkhd1l1* in hair cells compared to surrounding cells (Fig. 2c). At P2, translated *Pkhd1l1* mRNA was

much lower in the apical cochlea than in middle and basal regions. (Fig. 2d). Because there is temporal gradient of development at this age, the different levels of translated *Pkhd1l1* mRNA could be either tonotopic or developmental.

PKHD1L1 is a large, 4249-aa protein (Fig. 2a) with a strongly predicted signal sequence, which is likely to be largely extracellular. It has a single transmembrane domain, and a very small, 8-aa intracellular C-terminus. Its predicted domain structure includes 13 Ig-like, plexin, transcription factor (IPT) domains, which have an immunoglobulin-like fold and are similar to the EC domains of cadherins. IPT domains are usually found in cell surface receptors[30]. The 12 or more parallel beta helix repeats (PbH1) each have three beta strands in a circle; multiple PbH1 repeats usually stack to coil into a large helix. Proteins with PbH1 repeats often bind to polysaccharides[31]. PKHD1L1 has two G8 domains; these have 10 beta strands and an alpha helix, and are similar to, but larger than, the IPT domains[32].

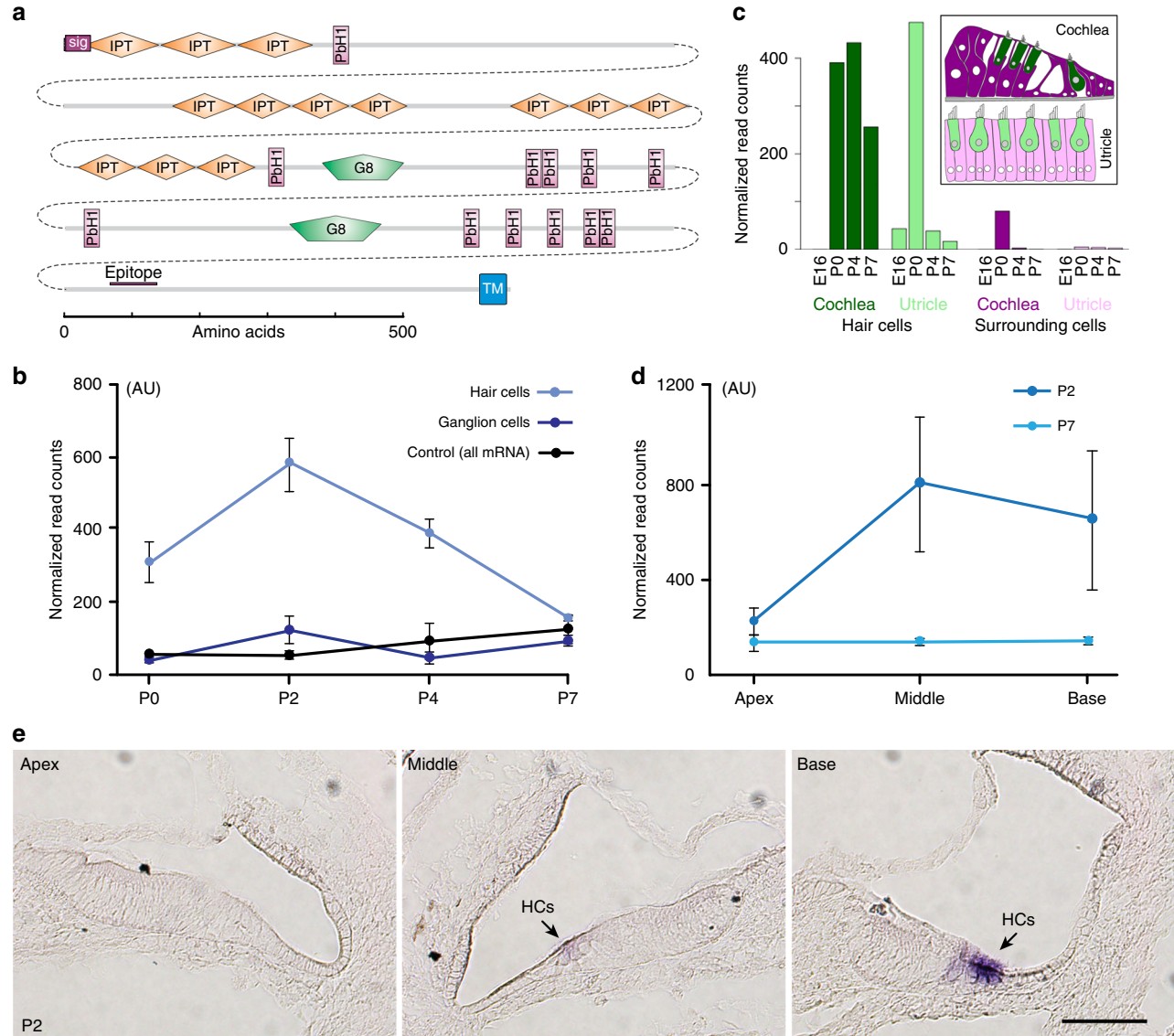

**Fig. 2** PKHD1L1 domain structure, transcription, and translation levels. **a** Domain structure of PKHD1L1. PKHD1L1 is 4249-aa long, with a single-transmembrane domain (TM) and an 8-aa intracellular C terminus. Shown also is the epitope for the commercial antibody used (NBP2-13765). **b** *Pkhd1l1* translation by hair cells and ganglion cells vs. control (all mRNA) for P0-P7 mice. **c** Relative *Pkhd1l1* mRNA levels in hair cells vs. surrounding cells in cochlea and utricle[29]. Inset: color coding of cell types. **d** *Pkhd1l1* translation by hair cells and ganglion cells vs. control (all mRNA) along the cochlea at P2. **e** RNA in situ hybridization showing an increase of *Pkhd1l1* mRNA levels, apex to base, in hair cells at P2. *Pkhd1l1* mRNA level is higher in OHCs than IHCs. In the basal turn, a faint *Pkhd1l1* mRNA signal is visible in the IHCs. Scale bar: 50 µm. Data shown as mean ± s.e.m. Source data are provided as a Source Data file

PKHD1L1 protein has not been previously shown to be expressed or function in hair cells. To confirm hair-cell specific expression of *Pkhd1l1*, we used in situ hybridization in the cochlea of P2 C57BL/6 mice (Fig. 2e). Apart from hair cells, we detected no visible signal in any other cell types in inner ear sections. Consistent with the *Pkhd1l1* translation data (Fig. 2d), the in situ images confirm higher levels of *Pkhd1l1* mRNA in the basal turn, with expression decreasing towards the apex.

**Generation of *Pkhd1l1^fl/fl* mouse line**. To understand the function of PKHD1L1, a floxed *Pkhd1l1* mouse line, with exon 10 flanked by *loxP* sites, was generated (Methods; Supplementary Fig. 1). Crossing with a *Cre*-expressing mouse line deletes exon 10 and causes a frameshift of the downstream coding sequence and a premature stop codon. *Pkhd1l1^fl/fl* mice were first crossed with a *Gfi1-Cre* mouse line, which expresses *Cre* in hair cells but no viable homozygotes were produced, suggesting embryonic lethality as a consequence of deletion in other cell types as well. We then crossed *Pkhd1l1^fl/fl* mice with *Atoh1-Cre* mice[33]. *Pkhd1l1^fl/fl*, *Atoh1-Cre^+* mice bred well, and appeared healthy. The genotyping primers of floxed *Pkhd1l1* allele were designed to detect flanking the upstream *loxP* site (Supplementary Fig. 1). As expected, the floxed allele generated a PCR product (624 bp) sized between the 700 and 500 bp ladders, whereas the wild-type allele generated a PCR product (438 bp) running slightly below the 500 bp ladder (Supplementary Fig. 1c). The ratio of offspring genotypes was consistent with Mendelian segregation.

**PKHD1L1 immunolocalization in hair cells**. With the knockout mouse, we could first validate an antibody and localize PKHD1L1 protein. We used a commercially available antibody (Novus Bio, Cat#NBP2-13765) raised against a human PKHD1L1 peptide, which corresponds to mouse PKHD1L1 aa3670–3739 (with a three amino-acid mismatch) downstream of the deleted exon (Fig. 2a). Anti-PKHD1L1 labeling detected no signal in stereocilia or the cell body of *Pkhd1l1^fl/fl*, *Atoh1-Cre^+* mice, validating the antibody specificity and further confirming a functional null *Pkhd1l1* allele (Supplementary Fig. 2). Antibody labeling then confirmed enrichment of PKHD1L1 in hair cells of the base compared to the apex (Fig. 3a–c). Strong fluorescence labeling in OHC stereocilia was detected at P2-P7 (Fig. 3d–f). Labeling intensity of bundles gradually decreased from P9 to P12 (Fig. 3g, h), and was no longer detected at P14 (Fig. 3i), suggesting downregulation of *Pkhd1l1*. IHC stereocilia bundles show lower intensity PKHD1L1 labeling overall (Fig. 3d–f).

We then used stimulated emission depletion (STED) microscopy to assess the subcellular localization of the PKHD1L1 (Fig. 3j). STED imaging indicates that PKHD1L1 is predominantly expressed at the upper half of the stereocilia. In addition, the PKHD1L1 signal is mostly located between the stereocilia, consistent with most of the protein being extracellular.

Surface specializations are differentially sensitive to exposure to the $Ca^{2+}$ chelator BAPTA and the protease subtilisin[4]. We evaluated the sensitivity of anti-PKHD1L1 staining to subtilisin and BAPTA by treating acutely dissected cochlear coils with subtilisin (Supplementary Fig. 3a–f) or BAPTA (Supplementary Fig. 3g–j) for 15 min at room temperature, followed by immunostaining. PKHD1L1 labeling is sensitive to subtilisin but not BAPTA treatment, as are both the stereociliary coat and the TM attachment crowns, suggesting that PKHD1L1 may participate in either of these structures (Table 1). Notably, the stereocilia coat is reported to be only partially sensitive to subtilisin treatment[4], suggesting it may be formed by a combination of proteins, both sensitive (like PKHD1L1) and resistant to subtilisin treatment.

**PKHD1L1 electron microscopic (EM) localization in hair cells**. To learn more about the location of PKHD1L1 in hair cells we utilized high-resolution EM localization methods. To test whether PKHD1L1 labeling is associated with any known surface specializations, we carried out anti-PKHD1L1 immunogold labeling using scanning electron microscopy (SEM), transmission electron microscopy (TEM), and focused-ion-beam SEM (FIB-SEM). In P4 OHC bundles, SEM revealed multiple anti-PKHD1L1 12 nm immunogold beads near the tips of stereocilia (Fig. 4a), whereas the nearby IHC bundles showed fewer beads (Supplementary Fig. 4). Presence of anti-PKHD1L1 gold beads at the surface of stereocilia is consistent with the prediction that PKHD1L1 is a largely extracellular membrane protein, and that the antibody recognizes its extracellular epitope. We further validated the anti-PKHD1L1 antibody in PKHD1L1-deficient cochleas, confirming its high specificity: only one bead (Fig. 4b, yellow arrow) was detected per 5–6 stereocilia bundles imaged.

We then processed immunogold-labeled samples for TEM imaging. Anti-PKHD1L1 10-nm gold beads were observed towards the tips of stereocilia and were often associated with the surface coat, which was seen as a dense, uniform fuzz on the membrane (Fig. 4c–h). As with SEM, gold beads were more often observed at the surface of OHC than IHC stereocilia (Fig. 4c, e–h). We also observed gold beads along the surface of the kinocilium (Fig. 4d). No gold beads were associated with the tip links or the transient lateral links connecting adjacent stereocilia.

To quantitatively evaluate the distribution of PKHD1L1 and its density at the surface of stereocilia, we collected three-dimensional data sets using FIB-SEM (Fig. 5) from P4 OHC stereocilia. In brief, FIB-SEM uses a focused gallium-ion beam to etch the surface in 15-nanometer steps, and SEM with backscatter detection to image the heavy metal counterstain and 10-nm gold beads at each freshly etched surface. Sequential milling and imaging generated serial EM data sets, which were further analyzed to generate 3D reconstructions of six hair-cell stereocilia bundles (see Methods). All bundles were collected from the same organ of Corti sample, and were located within close proximity, in the middle cochlear turn. Fig. 5a, b shows one such 3D-reconstructed bundle, with each yellow dot representing a gold bead observed at the surface of stereocilia (see also Supplementary Movie 1).

We used custom MATLAB data analysis software to generate average 3D distribution maps of the gold beads from the reconstructed bundles (see Methods). In brief, tall-, middle-, and short-row stereocilia, and kinocilia, were individually aligned with respect to the direction of their mechanosensitivity axis, and their height normalized to the tallest stereocilium within each row. Following normalization, the position of each gold particle was represented in a cumulative plot. The representative set of three stereocilia and the kinocilium was generated by displaying a small portion (~10%) of the surface, to represent an average ciliary surface within each row for all cilia analyzed (Fig. 5c). Thus, the surface map is a representation of the average shape of the stereocilia used in our data analysis.

These data ($n = $ ~200 cilia of each row from six OHC bundles; Fig. 5c) reveal that PKHD1L1 is localized toward the tips of stereocilia, with the highest density of the beads at the top ~300 nm of the cilia in all three rows. Gold beads formed a uniform ring around the tip of the tallest row of stereocilia, whereas the tips of second and third row stereocilia were labeled more on the negative side of their mechanosensitivity axis, with fewer beads between stereocilia (Fig. 5c, Supplementary Movie 2).

Next, we quantified the labeling density of different longitudinal segments of stereocilia and around the stereocilia circumference. The average height of stereocilia was 2.39 ± 0.16, 1.93 ± 0.14, and 1.24 ± 0.07 μm, whereas the average diameter was 157 ± 15, 182 ± 16, and 129 ± 13 nm for the tallest, middle, and

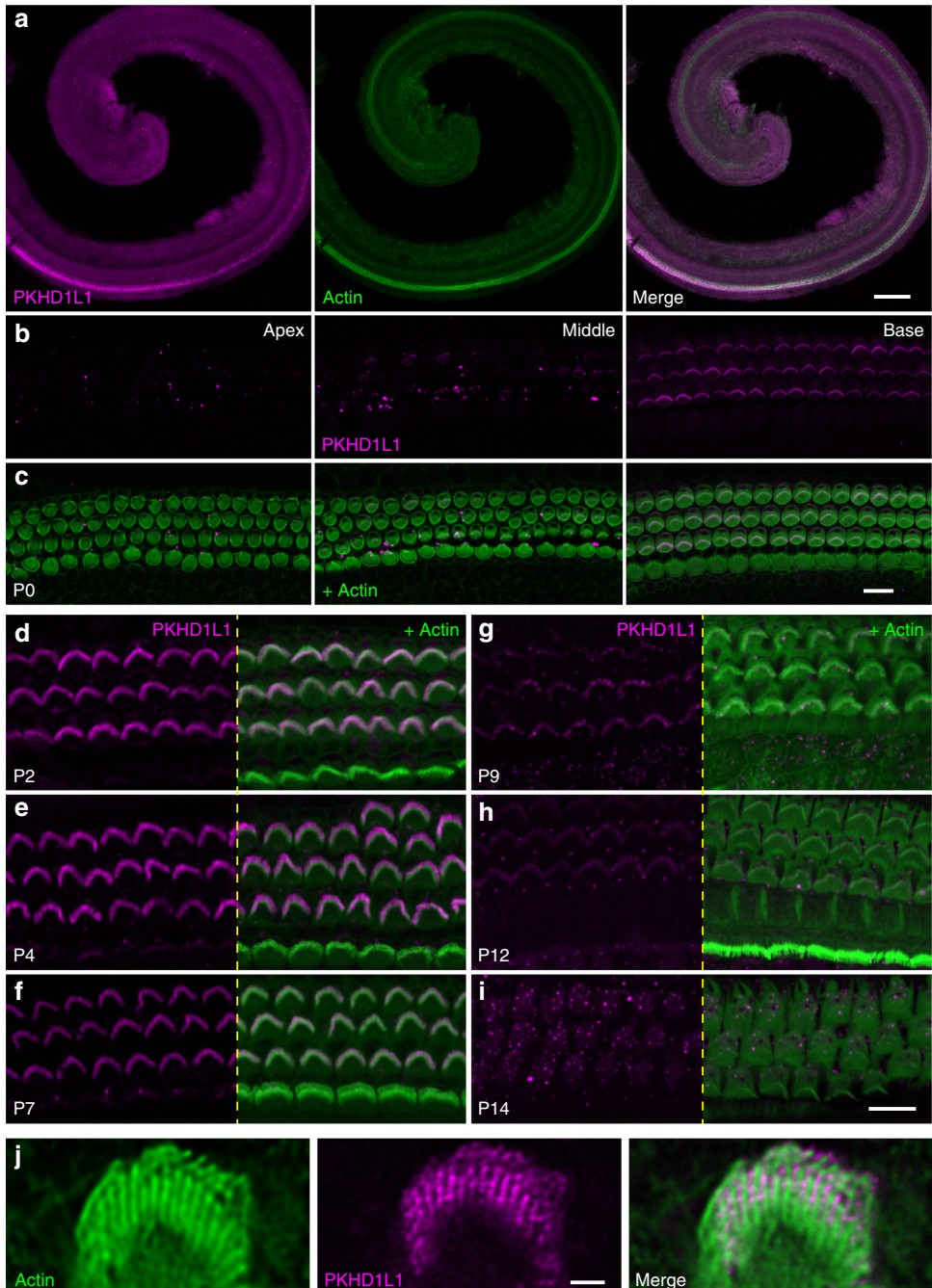

**Fig. 3** PKHD1L1 protein levels in hair cells of *Pkhd1l1fl/fl*, *Atoh1-Cre⁻* in early postnatal animals. **a** Confocal images of cochlear whole mounts with anti-PKHD1L1 labeling (magenta), and phalloidin (green; staining filamentous actin) at P0. Note the more abundant anti-PKHD1L1 signal in the base of the cochlea. **b**, **c** PKHD1L1 expression levels in P0 cochlea in three locations: apex (*left*), middle turn (*middle*), and base (*right*). **c** The phalloidin signal (green) is superimposed to visualize stereocilia bundles. **d–i** Strong anti-PKHD1L1 labeling is evident in OHC stereocilia bundles at P2–P7 **d–f**, decreasing from P9 to P12 **g**, **h**, and no longer detected with fluorescence within the bundles at P14 **i**. All images were collected from the mid-cochlear location. Anti-PKHD1L1 in magenta, phalloidin in green. **j** Stimulated Emission Depletion (STED) microscope imaging of PKHD1L1 labeling (magenta) on OHC stereocilia bundles at P7, combined with the phalloidin (green). Scale bars: **a**, 150 μm; **b–i**, 10 μm; **j**, 1 μm

shortest rows, respectively (mean ± SD). To quantify the labeling density per unit area, we divided each stereocilium into 200-nm segments along the length (12, 10, and 6 segments for the tallest, middle, and shortest rows, respectively). We further divided each segment into four radial sectors (anterior, posterior, and two lateral). The area of each segment, although of the same height, varied among the rows because of the difference in stereocilia diameter, resulting in 0.025, 0.028, and 0.021 μm² for the tallest, middle, and shortest rows respectively. Normalized labeling density (gold count per segment area, per cilium) is presented in Fig. 5d, where the color coding reflects the density values across the data set. For all three rows of stereocilia, gold bead density was significantly higher at the tips of stereocilia and enriched at the negative side of the tips for the middle and shorter row stereocilia, whereas more uniformly distributed around the tip of the tallest ones. Gold beads were also observed at the surfaces of kinocilia, although they were omitted from our analysis owing to a relatively low number of beads and kinocilia ($n = 6$).

**Table 1 Summary of the distribution and reagent sensitivity of the various cell surface specializations**

|  | Transient lateral links | Tip links | Ankle links | Horizontal top connectors | TM attachment crown | Apical surface coat | Stereociliary coat | PKHD1L1 |
|---|---|---|---|---|---|---|---|---|
| E17 | + | + | − | − | − | − | + |  |
| P0 | + | + | − | − | − | − | + | + |
| P2 | +/− | + | + | − | + | + | + | + |
| P5 | +/− | + | + | − | + | + | + | + |
| P9 | − | + | + | +/− | + | + | + | + |
| P12 | − | + | − | + | + | + | +/− | + |
| P14 | − | + | − | + | + | + | +/− | − |
| P19+ | − | + | − | + | + | − | − | − |
| Subtilisin | − | + | − | − | + | +/− | +/− | + |
| BAPTA | + | + | − | − | − | − | − | − |

Data from ref. [4] and this study

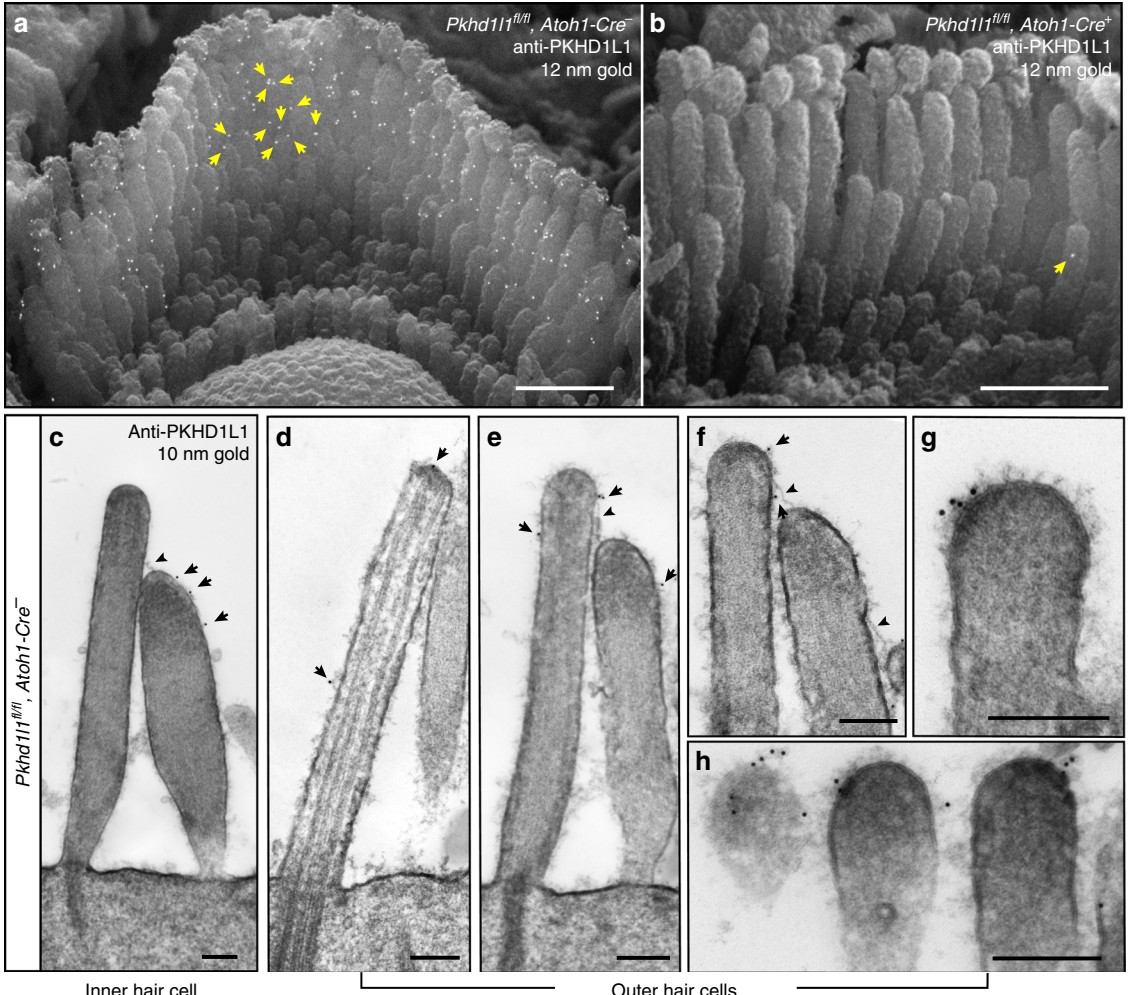

**Fig. 4** Immunogold localization of PKHD1L1 towards the tips of OHC and IHC stereocilia (P4). **a**, **b** SEM photomicrographs of P4 OHC stereocilia bundles. Multiple 12-nm gold beads in **a** (yellow arrows point to some gold beads) were detected on the surface of control *Pkhd1l1$^{fl/fl}$, Atoh1-Cre$^−$* OHCs, but not mutant *Pkhd1l1$^{fl/fl}$, Atoh1-Cre$^+$* hair cells, further validating the antibody specificity at the EM level in **b** (yellow arrow shows nonspecific label). **c–h** 10-nm gold beads were detected on TEM photomicrographs. Black arrows point to gold beads on the surface of IHC stereocilia **c**, OHC kinocilia **d**, and stereocilia **e–h**; black arrowheads point to the tip links **c**, **e**, **f**. Scale bars: **a–b**, 1 μm; **c–h**, 200 nm

**Little consequence of PKHD1L1 deletion for hair-cell morphology**. To understand the function of PKHD1L1, we examined the effect of PKHD1L1 deficiency on hair bundle morphology and overall organ of Corti morphology during bundle development. Previous studies revealed that PCDH15, involved in forming some stereocilia surface specializations, also contributes to the establishment of hair-cell's planar cell polarity (PCP), as mice lacking one of its isoforms show PCP defects[34]. We evaluated stereocilia bundle orientation in cochlear whole mounts, with actin stained with phalloidin to visualize stereocilia bundles.

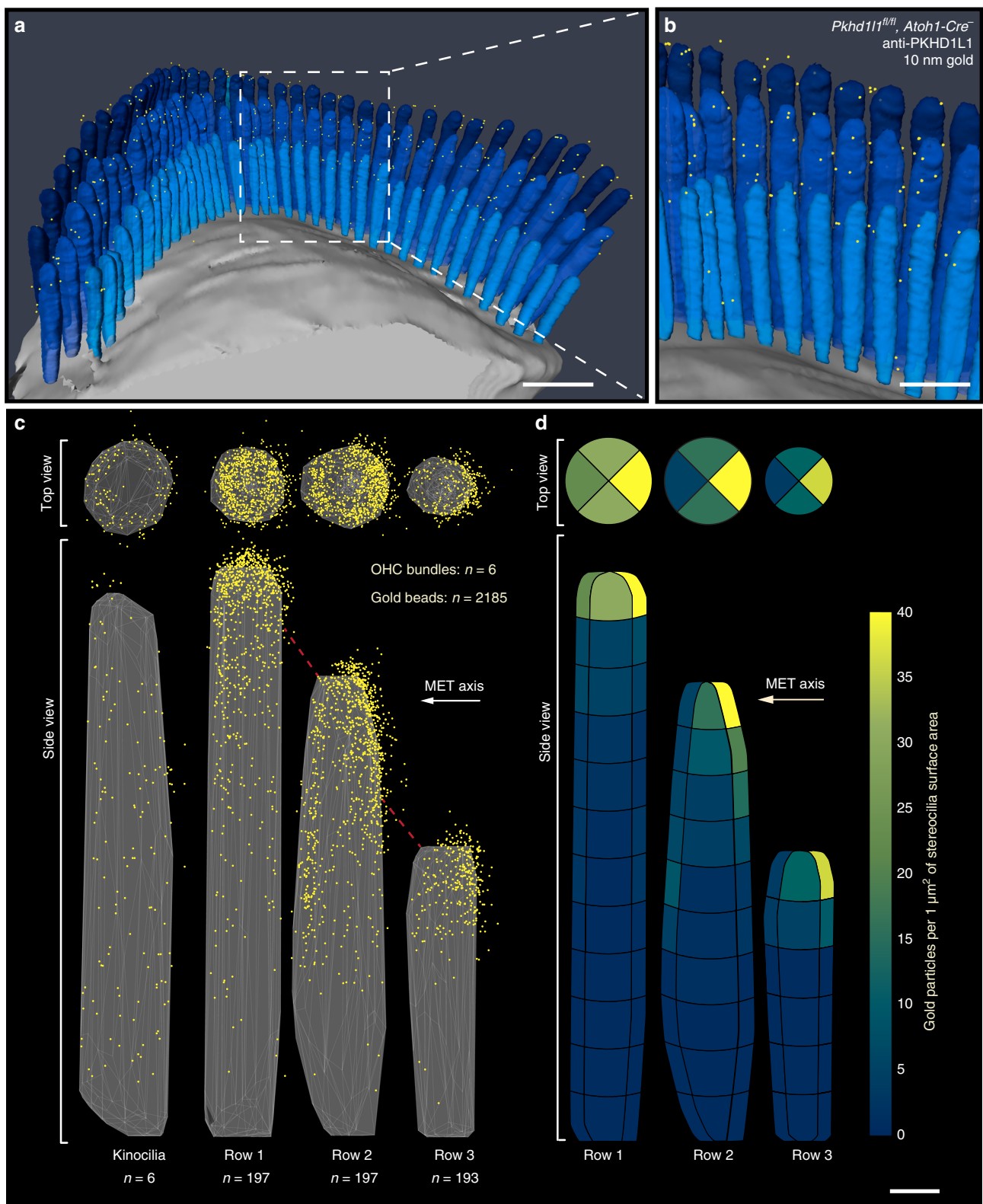

**Fig. 5** 3D localization of PKHD1L1 in P4 OHCs using immunogold FIB-SEM. **a**, **b** A 3D reconstruction of an anti-PKHD1L1 immunogold (10 nm)-labeled OHC stereocilia bundle (387 serial FIB-SEM cross-sections, at 15 nm milling step; yellow, gold beads; blue, stereocilia; gray, cell body). **c** A 3D cumulative distribution map of anti-PKHD1L1 immunogold beads generated with a custom MATLAB algorithm from six reconstructed OHC stereocilia bundles. Top and side views are shown for all three rows of stereocilia (yellow represents gold beads; gray triangles represent the stereociliary surface; red dashed lines represent tip links. Kinocilia (left) were excluded from further analysis due to a low number of observations. **d** Density of anti-PKHD1L1 immunogold beads (beads $\mu m^{-2}$ of stereociliary surface area). Same data as in **c**, calculated as an average density of beads within 12 (tallest row), 10 (middle row), or 6 (shortest row) equally tall segments of stereocilia. Each segment was further divided into four radial sectors, as seen in the top-view panel. Scale bars: **a**, 1 μm; **b**, 500 nm; **d**, 200 nm. Source data are provided as a Source Data file

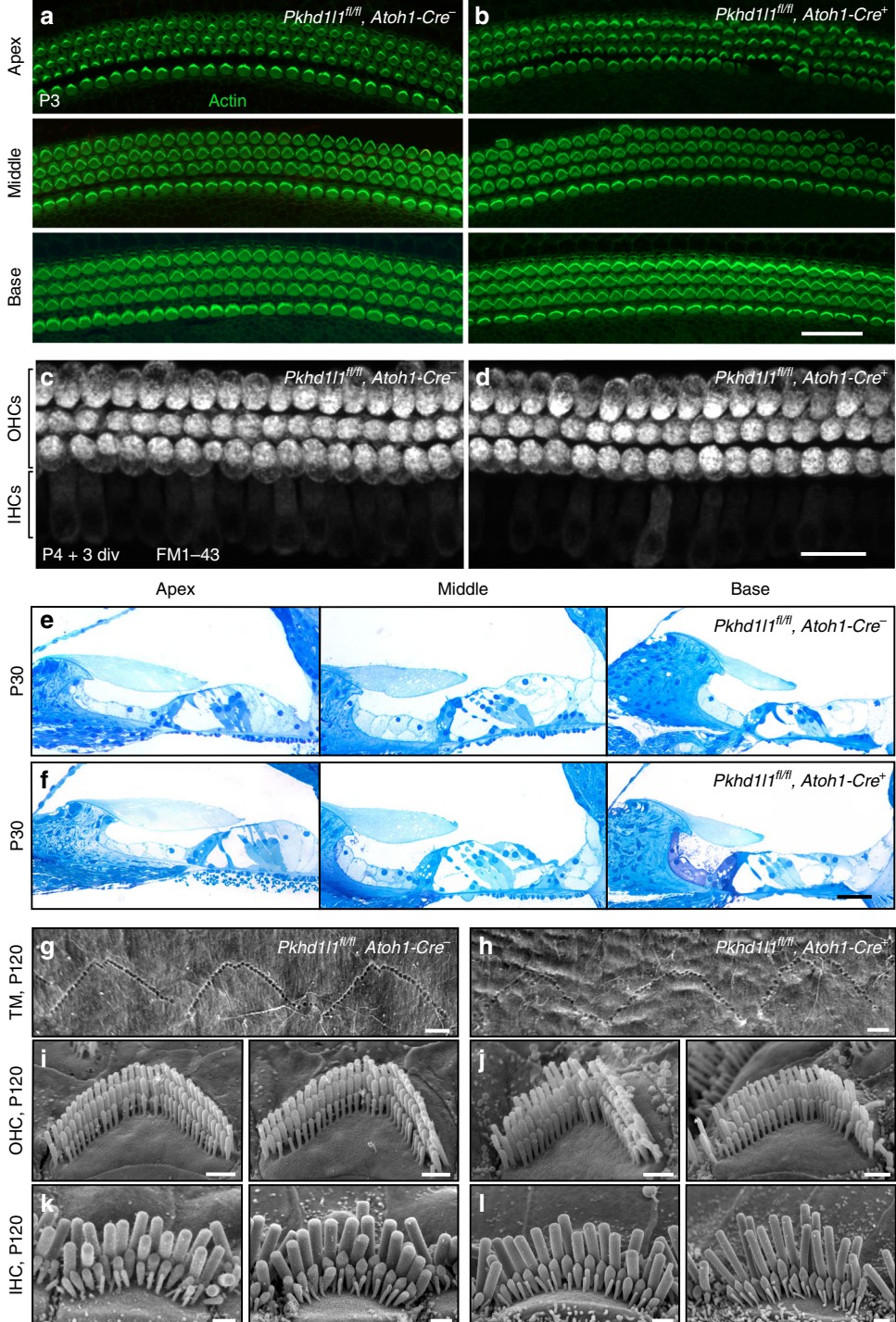

**Fig. 6** No gross deficit in bundle morphology or mechanotransduction in *Pkhd1l1^fl/fl^*, *Atoh1-Cre^+^* mice. **a**, **b** Hair-cell planar cell polarity appeared normal at P3 in the apical, middle, and basal turns of both mutant *Pkhd1l1^fl/fl^*, *Atoh1-Cre^+^* mice **b** and control *Pkhd1l1^fl/fl^*, *Atoh1-Cre^−^* littermates **a**. **c**, **d** PKHD1L1-deficient P4 + 3-days-in-vitro hair cells accumulate FM1-43. **e**, **f** Semithin plastic sections stained with methylene blue show no gross morphology deficit in the organ of Corti. In both groups at P30, the TM appears slightly retracted from the OHCs, likely an artifact of tissue preparation. **g**, **h** SEM photomicrographs of OHC stereocilia imprints on the surface of the TM. Basal turn, age P120. **i–l** SEM photomicrographs of *Pkhd1l1^fl/fl^*, *Atoh1-Cre^−^* **i**, **k**, and *Pkhd1l1^fl/fl^*, *Atoh1-Cre^+^* **j**, **l** IHC and OHC stereocilia bundles at P120. Scale bars: **a**, **b** 30 μm; **c**, **d** 20 μm; **e**, **f** 500 μm; **g–l** 1 μm

These showed no PCP defects in PKHD1L1-deficient hair cells at P3 (Fig. 6a, b). Hair-cell mechanotransduction was visually assessed by the brief application (30–60 s) of the small, positively charged lipophilic dye FM1-43[35], known to permeate hair cells through functional mechanotransduction channels. FM1-43 loading by PKHD1L1-deficient hair cells was similar to that in normal hearing littermates, suggesting no major deficiency of the hair-cell mechanotransduction complex (Fig. 6c, d).

To evaluate the anatomy of cochlear compartments throughout the cochlea, we analyzed semithin plastic sections (0.5 μm thick, stained with methylene blue) from the apical, middle, and basal cochlear coils of normal control, and PKHD1L1-deficient mice at

P30 (Fig. 6e, f). No gross anatomical deficit was detected by comparing control (upper) to knockout (lower) sections. In some sections we observed a slight retraction of the TM from OHC bundles in both groups, which is likely to be an artifact of sample preparation. The high expression levels of PKHD1L1 toward the tips of OHC stereocilia suggested that PKHD1L1 might be associated with stereocilia coupling to the TM. Previous studies have shown that in OHCs, the tallest row of stereocilia are tightly connected to the TM, forming indentations that correspond to their anchoring points[36]. To determine whether the TM attachment to OHCs is affected in PKHD1L1-deficient hair cells, we studied the TM at a higher magnification using SEM, previously used to visualize stereocilia imprints at the TM surface[25,37,38]. In the absence of PKHD1L1, as in control animals, three rows of V-shaped imprints were detected on the lower surface of the TM along the cochlea (Fig. 6g, h). In knockouts, their arrangement was not as highly preserved as in controls, but this is likely to be a consequence of the slightly altered V-shapes of OHC stereocilia bundles. Although the samples were collected from the same cochlear region, and processed through all steps in parallel, the TM surface of PKHD1L1-deficient mouse appears not as smooth as the TM surface of normal hearing littermates, possibly indicating that absence of PKHD1L1 could affect the structural integrity of the TM.

Finally, we evaluated the morphology of inner and outer hair-cell stereocilia bundles using SEM in young and adult mice, up to P120. Severe bundle disorder was not detected even at the latest age evaluated, P120 (Fig. 6i–l). However, some OHC bundles of PKHD1L1-deficient animals showed slightly disarrayed or even missing stereocilia (Fig. 6j). Such disorganization was not strongly pronounced nor systematic. We conclude that PKHD1L1-deficient stereocilia bundles develop normally and display no gross abnormalities as compared with those in control littermates.

**Progressive hearing loss in PKHD1L1-deficient mice**. Next, we evaluated hearing in mice lacking PKHD1L1 in hair cells. We measured Auditory Brainstem Response (ABR) and Distortion Product Otoacoustic Emissions (DPOAE) thresholds in response to pure tone stimuli, in *Pkhd1l1*$^{fl/fl}$, *Atoh1-Cre*$^-$ control, and *Pkhd1l1*$^{fl/fl}$, *Atoh1-Cre*$^+$ experimental animals. PKHD1L1-deficient mice show elevated thresholds in response to high-frequency tones (22.6 kHz, 32 kHz, and 45 kHz) as early as 3 weeks of age. The high-frequency hearing loss persisted and increasingly progressed in older animals, resulting in statistically significant threshold elevation at 16–32 kHz at 6 and 12 weeks of age. By 6 months, the latest age tested, *Pkhd1l1*$^{fl/fl}$, *Atoh1-Cre*$^+$ animals showed significantly elevated thresholds across all frequencies tested, often with no detectable thresholds at 80 dB SPL, the maximum level tested (Fig. 7a, b).

Overall, elevated ABR thresholds of *Pkhd1l1*$^{fl/fl}$, *Atoh1-Cre*$^+$ animals closely followed the DPOAE thresholds, suggesting the hearing phenotype is likely to arise primarily from an OHC deficit. The ABR peak 1, which represents the summed activity of the cochlear nerve (at 11.2 kHz, the most sensitive stimulus), was reduced, whereas the latency of ABR peak 1 and the width of ABR wave 1 were increased as compared with normal hearing littermates (Fig. 7c, d).

Because C57BL/6 J mice carry the *Cdh23*$^{ahl}$ mutation causing progressive hearing loss, and our animals were on a mixed C57BL/6 N/129S4/CBA genetic background, we confirmed that all animals tested were homozygous for *Cdh23*$^{ahl}$ and that the observed results did not arise from a mixed *Cdh23*$^{ahl}$ genotype. Our ABR and DPOAE results are consistent with recent phenotypic evaluations of two independent PKHD1L1 mutant mouse lines, in which disruption of PKHD1L1 decreased the

acoustic startle response (https://www.mousephenotype.org/data/genes/MGI:2183153).

**Surface coat at stereocilia tips in PKHD1L1-deficient mice**. In order to better visualize the surface specializations along the length of stereocilia, we used two different TEM sample staining protocols: either a brief 1% tannic acid treatment to highlight the links (Fig. 8a–c, i–k), or an overnight 5% tannic acid treatment to highlight the surface coat (Fig. 8d–g, l–n). As PKHD1L1 is expressed more in the basal turn of the cochlea, we focused on characterizing the surface coat within that region, at P4. Tip links, distinct ankle links and lateral links were clearly visible on the hair bundles of OHCs in both control and PKHD1L1-deficient hair cells (Fig. 8a–c, i–k). Similar to previous reports, in control OHCs electron-dense material was observed at the surface of stereocilia and the cuticular plate (Fig. 8d–h)[4,39]. Meanwhile, PKHD1L1-deficient hair cells showed a different pattern of cell coat distribution along the stereocilia. The dense coat overlaid the apical cell surface and the bottoms of the stereocilia, as in controls, but receded toward the upper ends of all rows of stereocilia, leaving the tips mostly smooth and free of any coating (Fig. 8l–o).

We then quantified the density of the stereociliary coat by generating average intensity plots across the membrane from outside to inside, in three stereociliary locations (Fig. 8p–r, lower panels; see Methods). In brief, we measured the density of the surface coat near the tips (100 nm from the top) of the tallest and middle row stereocilia (boxes in Fig. 8p, r, lower panels), and at the bottom third of the tallest stereocilium (box in Fig. 8q, lower panels). The average intensity of the tannic acid stain was measured across the membrane, background subtracted, and normalized to actin density within the stereocilium (Fig. 8p–r, upper panels). By subtracting the PKHD1L1-deficient values (dark blue traces) from PKHD1L1-wild-type values (black traces), the contribution of PKHD1L1 to the coat was measured (shaded light blue). The coat intensity at the tips of stereocilia was significantly lower ($p < 0.05$) in PKHD1L1-deficient OHCs compared with that in control littermates (Fig. 8s, upper graph), where most of the anti-PKHD1L1 labeling was observed (Figs. 4, 5). At the lower region of stereocilia in OHCs, in contrast, the surface coat intensity remained unchanged in PKHD1L1-deficient mice. The membrane intensity staining was also unchanged between the animal groups, serving as an internal control (Fig. 8s, lower graph), confirming that all other stereociliary structures (actin core, membranes) received similar levels of staining. In a separate set of experiments we used Ruthenium Red staining as previously described[40], which is reported to stain glycoproteins of the stereociliary coat. We detected no visible difference in the intensity of Ruthenium Red staining in PKHD1L1-deficient hair cells as compared with the control littermates.

In summary, our data suggest that PKHD1L1 (1) is expressed by hair cells from P0 to P12, and is predominantly localized to the tips of OHC stereocilia, (2) shows stronger labeling at the basal than apical turn of the cochlea at P0, and (3) is predominantly expressed in OHCs but less in IHCs. We found also that (4) mice lacking PKHD1L1 show progressive hearing loss, and (5) mice lacking PKHD1L1 lack the stereociliary coat at the tips of stereocilia. We conclude that PKHD1L1 is the major component of the upper surface coat of hair-cell stereocilia, and is essential for normal hearing especially at high frequencies.

## Discussion

The 3D localization of anti-PKHD1L1 gold beads at the surface of stereocilia, the absence of the coat at the tips of PKHD1L1-deficient mice, and PKHD1L1's sensitivity to subtilisin treatment,

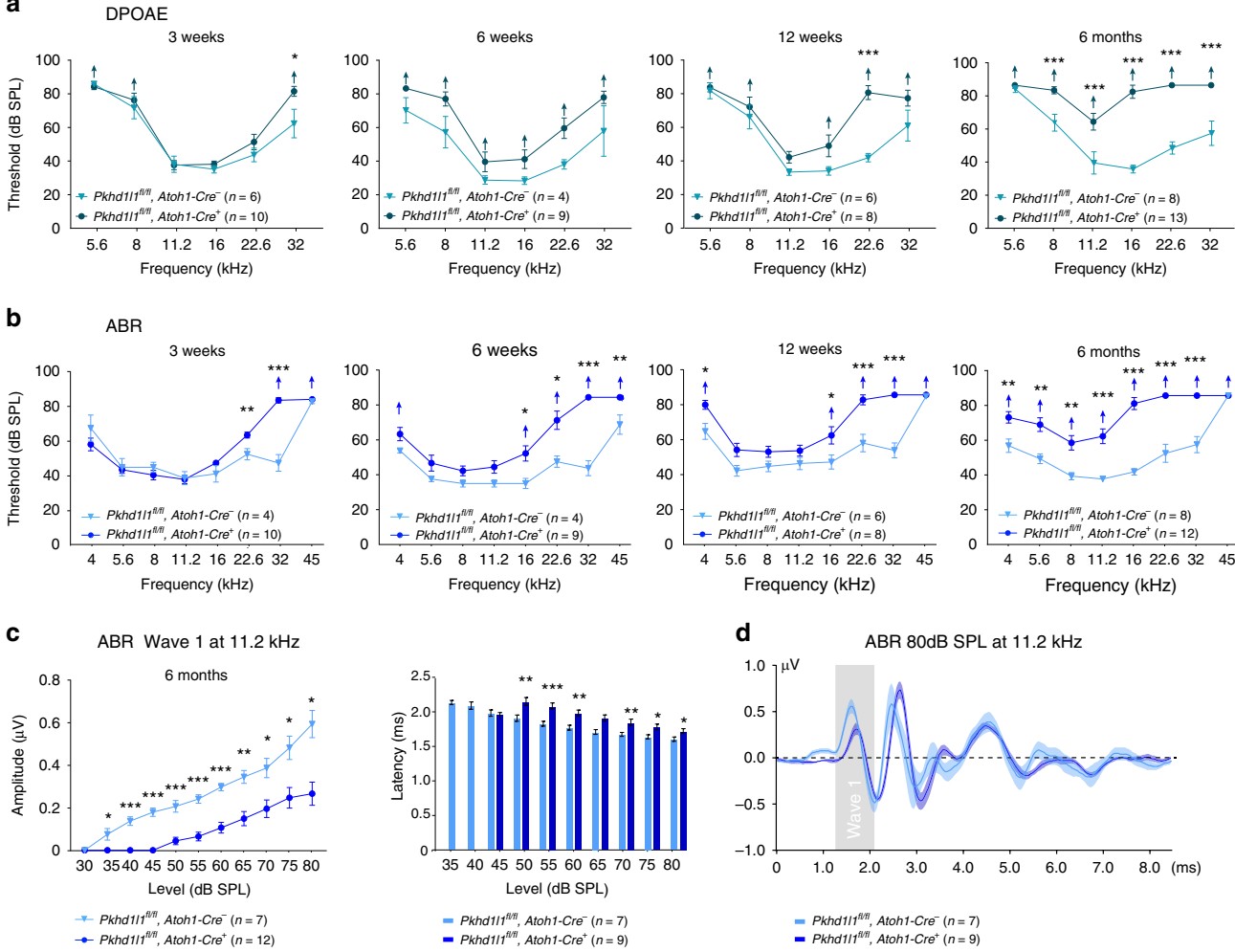

**Fig. 7** *Pkhd1l1fl/fl, Atoh1-Cre+* mice show progressive hearing loss. **a**, **b** DPOAE thresholds **a** and ABR thresholds **b** of *Pkhd1l1fl/fl, Atoh1-Cre−* and *Pkhd1l1fl/fl, Atoh1-Cre+* mice at 3, 6, 12 weeks, and 6 months show progressive hearing loss. Up-arrows indicate that even at the highest SPL test level (80 dB SPL), at least one mouse in the group had no detectable thresholds at that frequency. **c** Mean amplitude (left) and latency (right) as a function of sound intensity of 11.2 kHz pure tone pips in 6-month-old control *Pkhd1l1fl/fl, Atoh1-Cre−* (n = 7) and mutant *Pkhd1l1fl/fl, Atoh1-Cre+* (n = 12) mice. **d** Mean ABR waveforms in response to 11.2 kHz tone pips at 80 dB SPL from mutant (n = 9) and control (n = 7) mice. Data shown as mean ± s.e.m. *p < 0.05, **p < 0.001, ***p < 0.0001; by two-tailed Student's t test. Source data are provided as a Source Data file

together argue convincingly for involvement of PKHD1L1 in forming the surface coat. Notably, the stereocilia coat is reported to be only partially sensitive to subtilisin treatment[4], perhaps suggesting that a combination of proteins and their complexes, those both sensitive (like PKHD1L1) and resistant to subtilisin treatment, may be involved in forming the coat. This idea is also supported by the fact that PKHD1L1-deficient mice show a large decrease of the surface coat intensity only at the tips of stereocilia (Fig. 8) while the coat at the bases of stereocilia remains unaffected. The lower coat may be formed by a different set of proteins. These results further suggest that the surface coat may have a different function along the length of stereocilia, depending on its local composition.

Our immunogold labeling can only reflect a relative density of PKHD1L1 within different parts of the bundle, and does not measure the total number of molecules. Thus PKHD1L1 either might itself form the bulk of the surface coat at P4 at the tips, or facilitate the presence of other proteins that compose the coat.

What does the surface coat do? Some studies suggest the coat serves to attach the TM to the organ of Corti, and might promote stereocilia fusion following sound exposure or ototoxic drug administration[39]. The stereocilia coat was also suggested to contribute to bundle cohesion, allowing for stereocilia shear and contact while preventing membrane fusion, by creating a charged brush on stereocilia surfaces and enabling their sliding adhesion upon stimulation[13,26,41]. However it does not appear that PKHD1L1—less abundant in regions between stereocilia—is properly localized to mediate the sliding adhesion. Similarly, the timing of expression is inconsistent with PKHD1L1 mediating the horizontal top connectors that have been implicated in sliding adhesion[13]. As the coat disappears by P14-P19, it was also suggested to perform a transient role, mediating bundle maturation[4].

PLHD1L1 is a homolog of PKHD1, mutations of which cause autosomal–recessive polycystic kidney disease. Low levels of PKHD1L1 expression have been detected in primary immune cells, and although suggested to have a role in cellular immunity, its function is not understood[42]. The predicted domain structure of PKHD1L1 suggests that the ectodomain may bind other proteins (Fig. 2). The location of PKHD1L1 near the tips of OHC stereocilia suggests possible binding partners, but further studies are needed to evaluate possible interactions of PKHD1L1 with itself, with STRC and NPTN, with other TM components like

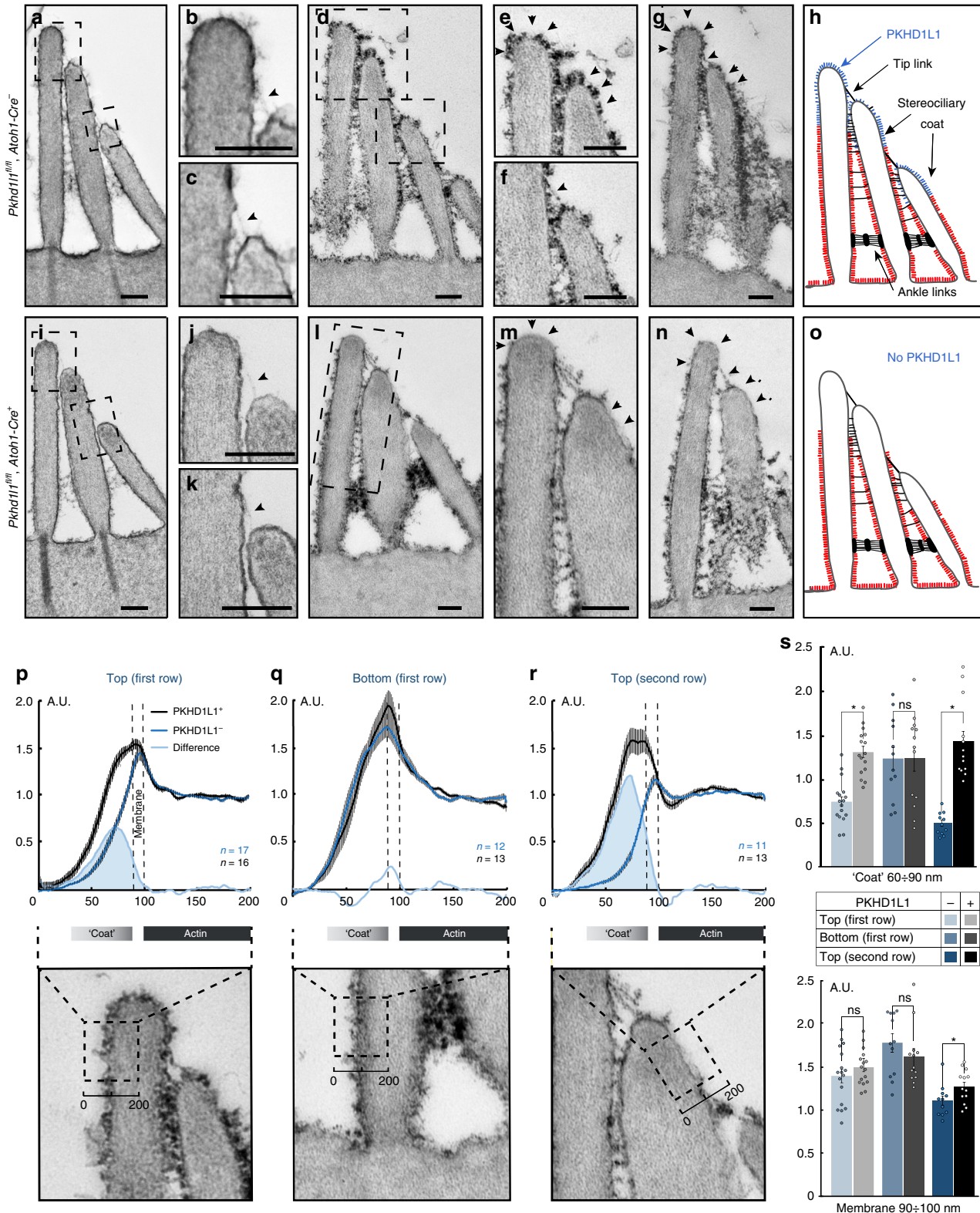

CEACAM16 and TECTA[25,43], or with other proteins that form stereocilia surface specializations. Such interactions would help our understanding of PKHD1L1 function.

During the first postnatal week, a rapidly changing array of transient links guide the bundle maturation. Growing amount of evidence suggest that link components are first produced in excess over the entire bundle surface. Then as development proceeds, they become progressively concentrated at the sites

where they form specific linkages, and excess protein is pruned away. This has been confirmed for the tip link proteins PCDH15 and CDH23, and the ankle-link antigen[4,15,17–19,44]. Similarly, TM attachment crowns are often seen on the shorter stereociliary rows at P2[4], but by P12 are mostly confined to the tips of the tallest stereocilia. At P4, PKHD1L1 is distributed towards the tops of all three rows of stereocilia, but may be further refined as the bundles mature. It is possible that PKHD1L1 expression at the

**Fig. 8** Reduced stereociliary coat at the tips of *Pkhd1l1*<sup>fl/fl</sup>, *Atoh1-Cre*[+] OHC stereocilia (P4). **a–c**, **i–k** TEM photomicrographs of adjacent stereocilia connected by the tip links (arrowhead) from control **a–c** and mutant littermates; **i–k** stained with 1% tannic acid. The presence of a tip link suggests the 80-nm section passes through the middle of stereocilium. **d–g**, **l–n** TEM photomicrographs of sections from basal turn samples stained with 5% tannic acid, and processed in parallel. Arrows point to differential staining intensity of the coat between control **e–g** and mutant **l–m** cells. **h**, **o** Schematic illustration of the coat distribution pattern in control **g** and mutant **n** bundles. Scale bars 200 nm. **p–r** Average intensity plots (upper panels) across the membrane, measured as shown in the lower panels. Black dashed boxes (200 nm × ~250 nm) outline the regions used to measure the staining intensity. All measurements were normalized to the intensity of actin staining. Black, control; dark blue, mutant; shaded light blue, the difference, indicating the PKHD1L1 contribution to the surface coat. Mean ± s.e.m. **s** Average coat and membrane intensities compared between the two groups (same data as **p–r**). Mean ± s.e.m., *n* = 11–17 sections per group from the basal cochlear region; *p < 0.05 by two-tailed Student's *t* test. Source data are provided as a Source Data file

---

surface of stereocilia decreases to levels undetectable by immunofluorescence labeling, as seen in Fig. 3i, but that some persists to adulthood. This might explain the discrepancy between the reduction in expression of PKHD1L1 by about P12, and the increase in hearing loss that is observed in PKHD1L1 mutants over several months.

Alternatively, the presence of PKHD1L1 in the first two postnatal weeks could be required for the proper localization of other proteins that remain in the stereocilia much longer and are needed for normal hearing in the adult. Animals that lack stereocilia link proteins (PCDH15, CDH23, USH2A, ADGRV1, PTPRQ) usually have severely damaged bundles and profound hearing loss (reviewed in ref. [3,7]), rather than the progressive hearing loss observed in PKHD1L1-deficient mice. The milder phenotype in PKHD1L1-deficient mice is more like that in mice with TM deficits, like STRC- and CEACAM16-deficient mice[45–47]. For instance, STRC-null mice have a progressive hearing loss with auditory thresholds above 10 kHz increasing from 25 dB at P15 to 60 dB at P60[10], whereas young CEACAM16-null mice show mild or unchanged ABRs, although they are reported to have altered TM structure[45]. On the other hand, PKHD1L1 is already present at the tips of stereocilia and may serve an important role at P2, when attachment crowns begin to form; both are sensitive to subtilisin, and both are localized to the tips of stereocilia from all rows at P4. Thus PKHD1L1 could participate in attachment crown assembly by forming immature crowns to initially secure the TM to the bundle, and its absence in early development could result in loose TM attachment persisting to adulthood.

Either way, this study reveals a new protein of hair-cell stereocilia, PKHD1L1, and clearly associates it with the surface coat of the upper part of OHC stereocilia. PKHD1L1-deficient mice have progressive hearing loss, indicating that this protein is essential for normal hearing. The PKHD1L1 model will allow more extensive characterization of the function of PKHD1L1, and by extension the upper surface coat of stereocilia.

## Methods

**Generation of knockout mice**. ES cell clones (Pkhd1l1<sup>tm2e(EUCOMM)Hmgu</sup>; clones HEPD0803-8-E04 and HEPD0803-8-H03) were purchased from the European Conditional Mouse Mutagenesis Program and the generated mice were identified by long template PCR that produces nucleotides flanking the short and long homologous recombination arms (Supplementary Fig. 1b). The frt-flanked Neo cassette was removed by crossing with a FLP deleter strain (The Jackson Laboratory, #003946). The conditional allele, which we refer to as *Pkhd1l1*<sup>fl/fl</sup>, contains two *loxP* sites flanking exon 10 of the *Pkhd1l1* gene. Cre recombinase removes exon 10 (71 bp), causing a frameshift and a premature stop codon. When crossed with the *Gfi1-Cre* mouse line (kindly provided by Dr. Lin Gan[28], the *Pkhd1l1*<sup>fl/fl</sup> mice produced no viable homozygous PKHD1L1-deleted progeny. We then crossed floxed mice with *Atoh1-Cre* mice (The Jackson Laboratory, B6.Cg-Tg(Atoh1-cre)1Bfri/J, #011104). The *Atoh1-Cre* recombinase disrupted the production of PKHD1L1 protein in hair cells (Supplementary Fig. 2). Genotyping of the *Pkhd1l1*<sup>fl/fl</sup> allele without the Neo cassette was performed by genomic PCR using the following primer pairs: GenoF2 (TGACACAACATACTGAGCAT) and 2r1 (GGAAACTCCTGTTGAAACAA). The floxed allele produced a 624-bp PCR band, whereas the wild-type allele produced a 438-bp band (Supplementary Fig. 1c). Genotyping primers used for the *Atoh1-Cre* allele were (1) Atoh1-Cre F:

ATCGGCCTCCTCCTCGTAGACAGC and (2) Atoh1-Cre R: GGATCCGCCGC ATAACCAGTGA. All experiments were carried out in compliance with ethical regulations and approved by the Animal Care Committees of the Harvard Medical School and Massachusetts Eye and Ear.

**In situ hybridization**. A PKHD1L1 cDNA fragment (bp7617-8320, 5′-atctgg-cagtctttgtacagca-3′, 5′-acaaacttagcactccagcctc-3′) produced by RT-PCR was subcloned into a pCRII-TOPO vector (ThermoFisher Scientific #K4610-20). cRNA sense and antisense probes tagged with DIG-11-UTP (Roche #11175025910) were produced using SP6/T7 RNA polymerase. Head blocks of P2 mice were cut as frozen sections at 10 μm thickness. In situ hybridization was performed as previously described[48]. Slides were mounted and imaged with an Olympus BX63 fluorescence microscope. The images were processed with ImageJ and Adobe Photoshop.

**Immunofluorescence labeling**. Mouse cochleas were dissected from P0-P28 mutant mice and their control littermates. Samples were fixed with 4% formaldehyde for 2 h, rinsed with PBS, and incubated in 0.1 M citrate buffer (pH 7.0) at 60 °C for 30 min for antigen retrieval. Whole-mount samples were permeabilized with 0.5% Triton X-100 for 30 min and blocked with 10% goat serum supplemented with 0.5% Triton X-100 for 30 min. Samples were then incubated with primary antibody overnight at 4 °C (1:200, Novus Bio #NBP2-13765), rinsed with PBS, and further incubated with donkey anti-rabbit (Alexa Fluor 568) secondary antibodies (1:200, ThermoFisher #A10042) supplemented with Alexa Fluor 488 phalloidin (1:100, ThermoFisher #A12379) for 6 h. Samples were then mounted with Prolong Gold antifade kit (Invitrogen), cured in the dark at room temperature overnight and imaged with an upright Olympus FluoView FV1000 confocal laser scanning microscope (60 × 1.42 NA objective).

**ABR and DPOAE**. ABRs and DPOAEs were recorded using an EPL Acoustic system[49] (for details see https://www.masseyeandear.org/research/eaton-peabody-laboratories/engineering-resources; Massachusetts Eye and Ear, Boston) in an acoustically and electrically shielded room. Adult mice (from P21 to P180) were anesthetized with intraperitoneal injection of a ketamine (100 mg kg⁻¹)-xylazine (10 mg kg⁻¹) cocktail and placed on a temperature-controlled heating pad set to 37 C for the duration of the experiment. The ear canal was exposed by making a small slit at the base of the pinna. ABRs were recorded using three subdermal needle electrodes: reference electrode in the scalp between the ears, recording electrode just behind the pinna near the slit, and ground electrode in the back near the tail.

For ABRs, 5-ms tone-pip stimuli with a 0.5 ms rise-fall time at frequencies from 5.6–45.2 kHz were delivered in alternating polarity at 30 s⁻¹. The response was amplified (×10 000), filtered (0.3–3 kHz), and averaged (×512) with a PC-based data acquisition system using the Cochlear function test suite software package (Massachusetts Eye and Ear, Boston, MA). Sound levels were incremented in 5-dB steps, from ~20 dB below threshold up to 80 dB SPL. ABR Peak Analysis software (v.1.1.1.9, unpublished; Massachusetts Eye and Ear, Boston) was used to determine the ABR thresholds and to measure peak amplitudes and latency. For traces with no detectablepeaks, the amplidute was reported as 0 μV, and the animal was excluded from the peak latency measurements. ABR thresholds were confirmed by visual examination as the lowest stimulus level in which a repeatable waveform could be observed. Up-arrows on some threshold points indicate that some animals in the group had no detectable thresholds at that frequency at 80 dB SPL, the maximum level routinely tested.

DPOAEs were recorded for primary tones (frequency ratio $f_2/f_1 = 1.2$, level ratio L1 = L2 + 10), where $f_2$ varied from 5.6 to 45.2 kHz in half-octave steps. Primary tones were swept in 5 dB steps from 10 to 80 dB SPL (for $f_2$). DPOAE threshold was determined from the average spectra as the $f_1$ level required to produce a DPOAE of -5-dB SPL.

**Subtilisin and BAPTA treatments**. The subtilisin and BAPTA treatment procedures were as previously described[4,19]. In brief, subtilisin (Sigma protease Type XIV) was freshly diluted to the working solution from a stock solution (×50) of 5 mg ml⁻¹ with Hanks' Balanced Salt Solution (HBSS) before use. BAPTA (5 mM)

was also freshly prepared in a calcium-free HBSS (pH = 7.2). The P4 cochleas were dissected and placed in either of the solutions for 15 min, then rinsed with HBSS three times, and subsequently fixed in 4% formaldehyde for 10 min. The cochleas were then immunolabeled as described above and imaged on a confocal microscope.

**Stimulated emission depletion microscopy.** The sample was processed as for immunofluorescence staining except that an Alexa Fluor 568 anti-rabbit secondary antibody was used. Alexa Fluor 488 phalloidin was used to counterstain the hair bundle. The samples were mounted in ProLong Diamond Antifade Mountant (Thermo Fisher #P36965), cured overnight at room temperature, and imaged using a Leica SP8 STED × 3 microscope. The images were further processed using Image J.

**FM1-43 loading.** Organ of Corti epithelia were acutely dissected from P4 mice in L-15 cell culture medium, and either mounted on coverslips using tungsten minute pins (WPI Inc.) or cultured for an additional 3 days in Dulbecco's Modified Eagle Medium, supplemented with 5% fetal bovine serum and $10 \, mg \, l^{-1}$ ampicillin. Following TM removal and medium aspiration, FM1-43 solution ($2 \, \mu M$ in L-15) was applied to the tissue for 30–60 s, then quickly aspirated. The explant was then quickly rinsed with L-15, and the excess dye quenched by a 0.2 mM solution of 4-sulphonato calix[8]arene, sodium salt (SCAS, Biotium) in L-15. The organ of Corti was then observed on an upright Olympus FV1000 confocal microscope, equipped with a 60 × 1.1 NA water-dipping objective lens.

**Scanning electron microscopy.** Organ of Corti explants were dissected at postnatal day 4 (P4) in L-15 medium and fixed with 2.5% glutaraldehyde in 0.1 M cacodylate buffer (pH 7.2), supplemented with 2 mM $CaCl_2$ for 1–2 h at room temperature. For older (P20–120) animals, temporal bones were fixed in 4% formaldehyde for 1 h at room temperature and decalcified in 5% ethylenediaminetetraacetic acid (pH 7.2–7.4) for 3–4 days at 4 °C. After unpeeling cochlear bone and removing the stria vascularis and TM, the cochlear coils were isolated, divided into apical, middle, and basal turns and postfixed with 2.5% glutaraldehyde in 0.1 M cacodylate buffer (pH 7.2), supplemented with 2 mM $CaCl_2$ for 1–2 h at room temperature. Samples were rinsed in 0.1 M cacodylate buffer (pH 7.2) and then in distilled water, dehydrated in an ascending series of ethanol, and critical-point dried from liquid $CO_2$ (Tousimis Autosamdri 815).

Samples were then mounted on aluminum stubs with carbon conductive tabs and were sputter-coated (EMS 300 T dual head sputter coater) with either 5-nm platinum (for conventional SEM) or 4.0-nm palladium (for immunogold-SEM) as previously described[19] and observed in field emission scanning electron microscope (Hitachi S-4800) or focused-ion beam (FIB) scanning electron microscope (FEI Helios 660) using secondary or backscatter electron detectors.

**TEM.** Organ of Corti explants were dissected at P4 in L-15 medium and fixed with 2.5% glutaraldehyde in 0.1 M cacodylate buffer (pH 7.2), supplemented with 2 mM $CaCl_2$. The fixative was supplemented with 1% tannic acid to visualize links (1–2 h at room temperature) or 5% tannic acid for stereocilia surface coat observation (applied overnight at 4 °C). Following a triple rinse in cacodylate buffer, they were postfixed with 1% osmium tetroxide/1.5% potassium ferrocyanide in 0.1 M cacodylate buffer for 2 h at room temperature in the dark. Explants were then washed three times in 0.1 Organ of Corti explants were dissected cacodylate buffer (pH 7.2), then briefly washed in distilled water, dehydrated in an ascending series of ethanol, equilibrated in propylene oxide and infiltrated and embedded in epoxy resin (Araldite 502/Embed-812 embedding media). Tissue pieces were positioned in molds and polymerized in the oven at 60 °C for 48 h.

The resin blocks were sectioned at 60–80 nm steps using a Reichert Ultracut S ultramicrotome. The sections were mounted on copper Formvar/Carbon-coated grids, stained with 2% uranyl acetate followed by lead citrate (Reynolds, 1963), and viewed with a JEOL 1200EX microscope operating at 80 kV. Images were captured with an Advanced Microscopy Techniques camera system at 3488 × 2580 pixel resolution.

**Stereocilia coat quantification.** To evaluate the density of the stereocilia surface coat, three distinct regions of stereocilia were analyzed. All samples were processed using the same staining procedures, in parallel. Images were collected using 10,000–20,000 magnification to minimize variability. For stereociliary tips, a box 100 nm below the tip of the stereocilium was used. For the measurements at the bottom of stereocilia, we used the lower third of the tallest stereocilium. The average pixel intensity was measured in ImageJ by placing a ~200 × 200 nm box perpendicular to the stereociliary membrane. The box was positioned such that the inner leaflet of the membrane always ran through the midline. Next, the intensity plot profile was measured as an average across the box. The background was measured from the same image within a region free of any cellular structures, and usually corresponded to the average value of the first ~20 nm of the intensity plot profile. Next, each intensity plot was background subtracted and normalized to the actin staining intensity (defined as the average value within 140–200 nm of the intensity plot measurement). Because the images were captured using different magnification, the resulting pixel size was either 0.77 nm, 0.94 nm, 1.26 nm, or 1.92

nm. We used linear regression to calculate the intensity values at 1 nm increments, and resulting individual traces were averaged across the group.

**Immunolabeling for electron microscopy.** Organ of Corti explants were dissected at P4 in L-15 medium, fixed with 4% formaldehyde in HBSS for 2 h at room temperature and washed in HBSS. Tectorial membranes were pulled away to expose the sensory epithelium after the fixation. Nonspecific binding sites were blocked by 10% goat serum in $Ca^{2+}$, $Mg^{2+}$-free HBSS for 2 h at room temperature. Samples were incubated overnight at 4 °C with primary antibodies diluted in blocking solution 1:200 (rabbit anti-PKHD1L1 antibody, NovusBio #NBP2-13765) followed by several rinses in $Ca^{2+}$, $Mg^{2+}$-free HBSS. Next, samples were incubated in blocking solution for 30 min at room temperature, and overnight at 4 °C with secondary antibody solution(1:30 in blocking solution). Immunogold Conjugate EM Goat F(ab′)2 anti-rabbit IgG:10 nm gold (BB International # 14216) secondary antibodies were used for immuno-TEM and immuno-FIB-SEM, and 12 nm Colloidal Gold AffiniPure Goat Anti-Rabbit IgG (H + L) (JAX Immunoresearch #111-205–144) for immuno-SEM. Following the secondary antibody application samples were rinsed in $Ca^{2+}$, $Mg^{2+}$-free HBSS (3×), postfixed in 2.5% glutaraldehyde in 0.1-M cacodylate buffer with 2 mM $CaCl_2$ for 1–2 h at room temperature and processed for SEM. Alternatively, samples were postfixed in 2.5% glutaraldehyde in 0.1 M cacodylate buffer (pH 7.2) supplemented with 2 mM $CaCl_2$ containing 1% tannic acid for 1–2 h at room temperature for TEM/FIB-SEM, then processed using the TEM sample preparation workflow described above.

**FIB SEM.** The resin blocks with immunogold-labeled tissue for FIB-SEM were prepared using the same protocol as the TEM sample workflow. The embedded specimen was exposed using a Reichert Ultracut S ultramicrotome equipped with diamond and trimming knives. The surfaces with oriented rows of inner or outer hair cells were confirmed by semithin sections observed on a light microscope following methylene blue staining.

Blocks containing the embedded tissue were then cut to a smaller size, and mounted on aluminum stubs using conductive carbon adhesive tabs, with the working surface facing upwards. All surfaces of the Araldite block, excluding the working surface were covered with a colloidal silver (or carbon) paint to increase sample conductivity, minimizing its charging, then sputter-coated with at least 5 nm of platinum.

Samples were observed on a FEI Helios 660 FIB-SEM microscope. Following local platinum deposition on the milling area, serial images of the block face were acquired by repeated cycles of sample milling and imaging using the Auto Slice & View G3 operating software (FEI) at a milling step of 10–20 nm.

**3D-structure reconstruction and volume analysis.** Image segmentation was carried out using Amira and Dragonfly image processing software packages.

Dragonfly was primarily used to align and filter the FIB-SEM image stacks. First, the image stacks were aligned by Maximization of Mutual Information. Aligned stacks were then subsequently filtered using edge-preserving and smoothing filters, such as Unsharp and Gaussian, respectively, to normalize the local contrast by simultaneously sharpening cellular membranes and gold beads. 3-D reconstruction of the image stacks was carried out with Amira, utilizing XY and XZ orthogonal slice views. With a size-7 brush, the outline of each cilium was traced on ~50 slices, skipping ~10–15 slices in between. The outlines of the cilium were then filled and the slices that were not reconstructed were completed using the Interpolation function. Most vivid gold beads were segmented using a size-5 brush on the XY orthogonal view. The pixel size varied from 1.95 to 2.60 nm in the data sets used in this study. The milling step for data sets analyzed in this study varied from 10 to 20 nm. This segmentation workflow produced stacks containing the kinocilium, three rows of stereocilia (manually assigned to the tallest, middle, and short rows), and the gold beads.

The post-segmentation analysis was performed using a custom MATLAB algorithm, which was developed at the Harvard Medical School Image and Data Analysis Core (IDAC) and is available upon request. The volume was resized (1:4) to speed up computation, and only 10% of border pixels (forming the stereociliary surface) were used for visualizations and some calculations. Once stereocilia surfaces were defined, each was analyzed to determine its ends and orientation, i.e., the coordinates for the tips and the tapers of each stereocilium. Each gold bead was reduced to a center of its centroid, with the centroids projected to the surface of the nearest stereocilium using the "k nearest neighbor" algorithm (with $k = 10$). Gold beads >100 nm away from the nearest surface were discarded from the analysis.

Once the gold beads were associated with their closest neighboring stereociliary surface, each stereocilium, along with the associated gold beads, was aligned in $X$, $Y$, and $Z$ axes. The height was normalized to the tallest stereocilium within each row. Next, based on the position of each stereocilium within the bundle, its azimuth was adjusted by rotating each cilium about its central axis, opening a bundle's V-shape to orient all cilia within the same plane of mechanosensitivity direction. Finally, all cilia from the same row were overlaid to generate the gold distribution map. The representative ciliary surface was rendered from overlaid surface coordinates representing individual cilia (10% of each surface), and triangulated within the closest neighboring coordinates.

To quantify the gold bead distribution along stereocilia, each stereocilium was divided into equal segments (~200 nm tall), four radial sectors each (anterior, posterior, and two lateral). The surface area of each sector was calculated using the known average diameter of stereocilia and the height of the sectors for each row. Normalized labeling density was then calculated as the gold count per segment area, per cilium for each stereociliary sector.

**Semithin sections and light microscopy**. Mice at P30 were anesthetized with isoflurane and euthanized by decapitation. The cochleas were then dissected in L-15 medium, and the oval and round windows opened. An additional small hole was made through the bone at the apical end of the cochlea. They were fixed in 2.5% glutaraldehyde/1% formaldehyde in sodium phosphate buffer pH 7.4 at room temperature for 4 h and further stored in fixative for ~12 h at 4 °C. After 3 washes in phosphate buffer, cochleas were postfixed with 1% osmium tetroxide aqueous solution for 2–4 h at room temperature in the dark. Next, they were washed three times in maleate buffer (pH 5.2) and stained with 1% uranyl acetate in maleate buffer for 2 h, washed again three times in maleate buffer and once in distilled water, dehydrated in an ascending series of ethanol, equilibrated in propylene oxide and infiltrated, and embedded in epoxy resin (Araldite 502/Embed-812 embedding media). Whole cochleas were then positioned in molds and polymerized in the oven at 60 °C for 48 h.

Semithin sections (0.5 μm) of the apical, middle and basal cochlear coils were cut from the cured plastic blocks using a Reichert Ultracut S ultramicrotome, then mounted on glass slides, stained with 1% methylene blue and imaged on a light microscope.

**Reporting summary**. Further information on research design is available in the Nature Research Reporting Summary linked to this article.

## Data Availability
The data sets generated or analyzed in the current study are available from the corresponding authors on reasonable request.

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

## Acknowledgements

We appreciate advice on scanning electron microscopy from Dr. William Fowle (Northeastern University) and the use of his SEM facility, on FIB-SEM microscopy from Drs. Austin Akey and Stephan Kraemer (Harvard University), and on TEM from Maria Ericsson, Elizabeth Benecchi, and Louise Trakimas (Harvard Medical School). We also thank the Neurobiology Imaging Facility (Harvard Medical School), which is supported by the National Institute of Neurological Disorders and Stroke under award number P30-NS072030. We thank the Dana-Farber/Harvard Cancer Center for the use of the Transgenic Mouse Core, which provided ES cell culture and injection service and is supported in part by NCI Cancer Center Support Grant P30-CA006516. We greatly appreciate assistance with laboratory management from Bruce Derfler and animal care from Yaqiao Li (Harvard Medical School). We thank Drs. Hunter Elliott and David Richmond (Harvard Medical School Image and Data Analysis Core) for advice with the MATLAB code. We thank Dr. M. Charles Liberman, Dr. Kirupa Suthakar, Ishmael Stefanov, Mike Ravicz, and Evan Foss for assistance with ABR data collection and analysis, and Dr. Brad Buran for developing the ABR peak-finder software. Research reported in this publication was supported by the National Institute on Deafness and Other Communication Disorders through grants R01-DC002881 and R01-DC016932 (to D.P.C.) and R01-DC017166 and R01-DC017166-S1 (to A.A.I.). This work was performed in part at the Harvard University Center for Nanoscale Systems (CNS), a member of the National Nanotechnology Coordinated Infrastructure Network (NNCI), which is supported by the National Science Foundation under NSF award 1541959. We especially appreciate institutional support to Harvard Medical School from the Bertarelli Foundation.

## Author contributions

X.W. carried out the Ribotag analysis, identified PKHD1L1 as a candidate stereocilium protein, constructed the PKHD1L1 knockout mouse, carried out initial characterization of the phenotype, performed fluorescence microscopy, analyzed the data, and helped write the paper. M.V.I. did fluorescence, TEM, SEM, and FIB-SEM microscopy, analyzed the data, created figures and movies, and helped write the paper. H.A.J. analyzed FIB-SEM data. M.C. wrote custom MATLAB code for quantifying and plotting bead density. A.A.I. carried out SEM and FIB-SEM microscopy, analyzed the data, oversaw the project, and helped write the paper. D.P.C. analyzed the data, oversaw the project, and helped write the paper.

## Additional information

**Competing interests:** The authors declare no competing interests.

**Peer Review Information:** *Nature Communications* thanks Joseph Santos-Sacchi, and the other, anonymous, reviewer(s) for their contribution to the peer review of this work. Peer reviewer reports are available.

