## [Peer Review File · Nature Communications]

Reviewers' comments:

Reviewer #1 (Remarks to the Author):

The manuscript from Wu et al. is a technically superb characterization of a key component of the outer hair cell's stereocilia coat, PLHD1L1. This protein is not essential for forming hair cells, stereocilia, or the final functional organ of Corti, but it does appear to be required for long-term maintenance as demonstrated by progressive hearing loss. Thus the manuscript is an important contribution to hair-cell biology.

While the manuscript could be seen to be of interest only to hair cell specialists, the one technical contribution here is an extremely robust immuno-FIB-SEM characterization of PLHD1L1 labeling on stereocilia. The methods are state-of-the-art and the presentation is superb.

Major comments

1. There is a significant discrepancy—which could just be interesting biology, not a technical problem—in that expression seems to cease by P12, yet the phenotype develops slowly over months. The slow developing phenotype would argue that the protein is present for a long time in controls. I realize that it is well outside the domain of this manuscript to measure the turnover of the cell coat or PLHD1L1 over time, but it would be worthwhile to look at protein expression and the cell coat in significantly older animals.

2. There is much discussion about the possibility of PLHD1L1 forming interactions with other cell surface proteins of the stereocilia. However, there is no evidence presented of such interactions and in fact the distribution shown from FIB-SEM suggests that PLHD1L1 is excluded from regions where stereocilium-stereocilium connections might form. Moreover, SEM of organ of Corti or other hair cell organs often shows substantial extracellular filamentous material—perhaps that is what PLHD1L1 binds to. Please discuss the lack of evidence for stereocilium-stereocilium connections including PLHD1L1.

Other comments

3. Page 2, line 35. The structure of this sentence makes the “all are encoded by genes...” ambiguous—does that modify “molecular components of many links” or the list of specific genes, which is not comprehensive?

4. Page 4, line 15. Needs an “and” between USH2A and ADGRV1.

5. Page 4, lines 16-18. Three examples of “contribute” in three sequential sentences.

6. Page 4, lines 23-24. This is the Introduction, so it is best to cite examples of transient link characterization and repulsive coat characterization.

7. Page 4, line 37. Add “...at their tips.”

8. Page 5, line 17. You can't conclude from these data that there is an apical to base gradient. It looks here like there is a peak in the mid-cochlear region.

9. Page 5, paragraph starting with line 20. Don't underline the letters that contribute to the abbreviations. The reader is perfectly capable of figuring this out and the underlining is distracting.

10. Page 7, lines 1-2. You have previously mentioned the source of the Gfi-Cre mouse, no need to

again.

11. Page 7, lines 9-11. Don't mention all of the genotypes unless you actually include the data to show Mendelian inheritance. It is all right just to state that, however.

12. Page 7, lines 23-24. Why do IHCs show lower labeling?

13. Page 7, lines 27-28. This point is not clear from the figure.

14. Page 8, Fig. 3. Our colorblind colleagues are pleased when you use magenta/green instead of red/green.

15. Page 14, line 30. Here the control is said to be $Pkhd111^{fl/+}; Atoh1-Cre^{-}$, but the figure says $Pkhd111^{fl/fl}; Atoh1-Cre^{-}$. Please correct one of them.

Reviewer #2 (Remarks to the Author):

This well-done manuscript focuses on one molecular component (PKHD1L1) of the apical surface coat of mammalian hair cells, principally located near stereociliar tips of OHCs. The surface coat was originally described by Santi's group in the 1980s. The authors provide exhaustive evidence on its identification, location and effects of its deletion on hearing in mice. Surprisingly, no morphological changes are apparent in the KO, but a developmental hearing loss restricted mainly to high frequencies is found. The authors conclude that PKHD1L1 is required for normal hearing.

I have some questions and comments.

Is there a true gradient of PKHD1L1 along the frequency axis? From Fig 3., it appears that it is rather abruptly present in the base.

What is the tannic acid result in the low frequency region of the cochlea? Is the surface coat fully normal in the KO? Such a result might indicate the existence of distinct components in different cochlear regions. Does the low frequency coat alter when low frequency hearing loss becomes evident at advanced ages? Are there other candidates for surface coat proteins that arise from your database?

Explain why Ruthenium Red observations were negative.

I have a concern regarding the conclusion that the protein is required for normal hearing. It seems to me that a true requirement when removed (i.e. in the KO mouse) would be evidenced across frequencies and early in life. Another more likely role for PKHD1L1 could be to limit deleterious effects on hearing during aging. It is well known that susceptibility to hearing loss during aging (or other insults, eg. noise exposure) may be augmented by combination effects (e.g., ototoxic drugs). What was the acoustic environment like during rearing of the animals? It is not uncommon for vivariums to have significant background noise. Perhaps the KO is more susceptible to these exposures? Have you tested whether these animals are more susceptible to the presentation of deleterious acoustic exposures? If so, PKHD1L1 would not be required for normal hearing, but would help to maintain normal hearing, especially in the high frequencies. This would be a significant finding, and would avoid speculation.

J. Santos-Sacchi

Reviewer #3 (Remarks to the Author):

General comments:

This outstanding manuscript presents results on a new protein, PKHD1L1, found in the coating of the upper portion of cochlear hair cell stereocilia. PKHD1L1 is expressed transiently during the first two weeks or so of development, but when absent, the animal becomes progressively deaf. The protein is novel in the ear and is named for its similarity to a polycystic kidney and hepatic disease protein, but it is of unknown function and molecular identity. The authors hypothesize that this extracellular protein may be involved in linking to the many other proteins that participate in forming the external coating and various stereociliar links and they suggest the intriguing function of allowing for the gliding action of stereocilia past each other and the facilitation of bundle movement.

Many, many approaches are used to localize this novel protein and its RNA on stereocilia and to demonstrate its function in both wild type and null mice. The data figures are each outstanding, with great attention to detail, and the supplementary video is very well-done, among the best that I have seen in this area. I have no major concerns.

Specific minor comments:

P. 5, L. 20. A minor typo. I believe that the sentence should read "PKHD1L1 is a large, 4249-aa protein..."

P. 12, L. 1. I was a little confused by the statement of number of stereocilia per row. Surely there aren't 200 cilia per row? I count maybe 40 stereocilia per row in the SEMs of the P120 OHCs and fewer in the IHCs.

P. 14, L. 15. Duly noted about the wrinkling on the TM in the nulls, but I wonder if there was any gold particle labeling for PKHD1L1 on the TM in the wild type? One supposes that this is a future direction, when examining interactions with TM proteins such as CEACAM16 and TECTA

P. 21, L. 1. Another minor typo. I suggest "...were cut as frozen sections at 10 um thickness."

References:

P. 33, L. 33-34. I suggest not using Title Case for this article title if it is not being used elsewhere.

Reviewers' comments:

Reviewer #1 (Remarks to the Author):

The manuscript from Wu et al. is a technically superb characterization of a key component of the outer hair cell's stereocilia coat, PLHD1L1. This protein is not essential for forming hair cells, stereocilia, or the final functional organ of Corti, but it does appear to be required for long-term maintenance as demonstrated by progressive hearing loss. Thus the manuscript is an important contribution to hair-cell biology.

While the manuscript could be seen to be of interest only to hair cell specialists, the one technical contribution here is an extremely robust immuno-FIB-SEM characterization of PLHD1L1 labeling on stereocilia. The methods are state-of-the-art and the presentation is superb.

Major comments

1. There is a significant discrepancy—which could just be interesting biology, not a technical problem—in that expression seems to cease by P12, yet the phenotype develops slowly over months. The slow developing phenotype would argue that the protein is present for a long time in controls. I realize that it is well outside the domain of this manuscript to measure the turnover of the cell coat or PLHD1L1 over time, but it would be worthwhile to look at protein expression and the cell coat in significantly older animals.

This is an interesting phenomenon (also noted by the reviewer #2), which we don't fully understand at this time. PKHD1L1 might still be present at the surface of stereocilia at later ages, although at levels that cannot be detected by conventional fluorescence labeling techniques. Alternatively, PKHD1L1 might carry out a mostly developmental role, and its absence results in higher susceptibility to ambient noise or dysregulated stereocilia bundle maintenance. This is an ongoing work in the laboratory of Dr. Indzhukulian, and we hope a follow up study could reveal the answer. The Discussion has been edited to highlight the two most likely explanations.

2. There is much discussion about the possibility of PLHD1L1 forming interactions with other cell surface proteins of the stereocilia. However, there is no evidence presented of such interactions and in fact the distribution shown from FIB-SEM suggests that PLHD1L1 is excluded from regions where stereocilium-stereocilium connections might form. Moreover, SEM of organ of Corti or other hair cell organs often shows substantial extracellular filamentous material—perhaps that is what PLHD1L1 binds to. Please discuss the lack of evidence for stereocilium-stereocilium connections including PLHD1L1.

It is true that we have no direct evidence of PKHD1L1 mediating such interactions, and have modified the Discussion to acknowledge that this possibility is less likely.

Other comments

3. Page 2, line 35. The structure of this sentence makes the “all are encoded by genes...” ambiguous—

does that modify “molecular components of many links” or the list of specific genes, which is not comprehensive?

Thanks; the wording been modified to indicate proteins rather than links.

4. Page 4, line 15. Needs an “and” between USH2A and ADGRV1.

This has been corrected.

5. Page 4, lines 16-18. Three examples of “contribute” in three sequential sentences.

These have been replaced.

6. Page 4, lines 23-24. This is the Introduction, so it is best to cite examples of transient link characterization and repulsive coat characterization.

We have included appropriate references.

7. Page 4, line 37. Add “...at their tips.”

This has been corrected.

8. Page 5, line 17. You can’t conclude from these data that there is an apical to base gradient. It looks here like there is a peak in the mid-cochlear region.

We agree, and have modified the wording to avoid the claim of a gradient.

9. Page 5, paragraph starting with line 20. Don’t underline the letters that contribute to the abbreviations. The reader is perfectly capable of figuring this out and the underlining is distracting.

The underlining has been removed.

10. Page 7, lines 1-2. You have previously mentioned the source of the Gfi-Cre mouse, no need to again.

The citation has been removed.

11. Page 7, lines 9-11. Don’t mention all of the genotypes unless you actually include the data to show Mendelian inheritance. It is all right just to state that, however.

Thank you for the suggestion.

12. Page 7, lines 23-24. Why do IHCs show lower labeling?

We don't know, and it is difficult to speculate without more knowledge of PKHD1L1's specific function. It is consistent with the idea that PKHD1L1 participates in the attachment of stereocilia to the TM, or to the development of that attachment, but this is perhaps too speculative for the manuscript.

13. Page 7, lines 27-28. This point is not clear from the figure.

We have magnified the figure to that the location of PKHD1L1 between stereocilia is more apparent.

14. Page 8, Fig. 3. Our colorblind colleagues are pleased when you use magenta/green instead of red/green.

We have changed colors throughout.

15. Page 14, line 30. Here the control is said to be $Pkhd1l1^{fl/+}; Atoh1-Cre^{-}$, but the figure says $Pkhd1l1^{fl/fl}; Atoh1-Cre^{-}$. Please correct one of them.

The figure genotype has been verified, and is $Pkhd1l1^{fl/fl}, Atoh1-Cre^{-}$.

Reviewer #2 (Remarks to the Author):

This well-done manuscript focuses on one molecular component (PKHD1L1) of the apical surface coat of mammalian hair cells, principally located near stereociliar tips of OHCs. The surface coat was originally described by Santi's group in the 1980s. The authors provide exhaustive evidence on its identification, location and effects of its deletion on hearing in mice. Surprisingly, no morphological changes are apparent in the KO, but a developmental hearing loss restricted mainly to high frequencies is found. The authors conclude that PKHD1L1 is required for normal hearing.

I have some questions and comments.

Is there a true gradient of PKHD1L1 along the frequency axis? From Fig 3., it appears that it is rather abruptly present in the base.

The legend to Figure 3 has been modified to include a statement highlighting that Panels d-i represent PKHD1L1 labeling in mid-cochlear location.

What is the tannic acid result in the low frequency region of the cochlea? Is the surface coat fully normal in the KO? Such a result might indicate the existence of distinct components in different cochlear regions. Does the low frequency coat alter when low frequency hearing loss becomes evident at advanced ages?

These are all good questions currently being investigated in the Indhzykulian laboratory, but they are beyond the scope of this first report on PKHD1L1. Here, we have intentionally limited our study to characterizing PKHD1L1 distribution to just one age— postnatal day 4—due to a very high complexity of

most of our electron microscopy experiments. Unfortunately, a simple evaluation of the surface coat at the low frequency region of the cochlea at P4 would not yield definitive answers, because there is a developmental gradient (apical vs. basal hair cell maturation) at that age. A more comprehensive study across three cochlear locations and several developmental stages is being carried out. To complete it at the level of detail used in this study at P4 would take much more time and unreasonably delay the publication of the paper.

Are there other candidates for surface coat proteins that arise from your database?

This is the only protein that we know of that forms the surface coat. Our search through the database has identified several other proteins that met our search criteria (over 1500aa, at least one transmembrane domain, and long extracellular domain), but we refer to these hits as “potential surface specializations;” we don’t have evidence that they are “surface coat” components. For instance, several proteins forming known surface specializations (STRC, PTPRQ, USH2A, CDH23, PCDH15, and ADGRV1) also met these search criteria, but are unlikely to be contributing to the surface coat in any major capacity.

Explain why Ruthenium Red observations were negative.

As a cationic dye, Ruthenium red has very high affinity for acidic sulfated residues and has been used especially for visualization of glycosaminoglycan chains of proteoglycans (Luft, J.H. 1971; Blanquet, P.R. 1976). Tannic acid contains phenolic and carboxyl groups and, when used in combination with glutaraldehyde and osmium tetroxide, interacts with many different types of proteins, glycoproteins, and other components, regardless of their electrical charge (Birembaut et al. 1982; Hayat 1989). Thus it is possible either that the negative cell surface charges which are responsible for a Ruthenium Red reaction are not fully developed at P4 or that PKHD1L1 does not contribute much to the formation of acidic glyco-conjugates along the stereocilia surface at that age.

I have a concern regarding the conclusion that the protein is required for normal hearing. It seems to me that a true requirement when removed (i.e. in the KO mouse) would be evidenced across frequencies and early in life. Another more likely role for PKHD1L1 could be to limit deleterious effects on hearing during aging. It is well known that susceptibility to hearing loss during aging (or other insults, eg. noise exposure) may be augmented by combination effects (e.g., ototoxic drugs). What was the acoustic environment like during rearing of the animals? It is not uncommon for vivariums to have significant background noise. Perhaps the KO is more susceptible to these exposures? Have you tested whether these animals are more susceptible to the presentation of deleterious acoustic exposures? If so, PKHD1L1 would not be required for normal hearing, but would help to maintain normal hearing, especially in the high frequencies. This would be a significant finding, and would avoid speculation.

We agree with this line of reasoning, but argue that the conditions in our animal facility could be referred to as “normal” as they were used for control animals; hence the phenotype the mutants develop with no *intentional* noise damage could be described as loss of “normal” hearing. We considered removing the word “normal” from the title, but that would suggest that PKHD1L1 is required for all hearing, which is not the case. Normal thus refers to the absence of progressive ABR threshold

shift in our rearing conditions, which are typical for laboratory mice. In order to test whether PKHD1L1 is required for normal hearing by the standards of mice in the wild, the animals would have to be raised in a very quiet environment for up to 6 months, which may be very difficult.

We agree that it is interesting to determine whether PKHD1L1-deficient mice are more susceptible to noise trauma, and we will address this in future work. For instance, noise trauma that results in reversible hearing loss (“temporary threshold shift”) in control animals could yield a more severe outcome in PKHD1L1-deficient mice and produce a permanent threshold shift.

Reviewer #3 (Remarks to the Author):

General comments:

This outstanding manuscript presents results on a new protein, PKHD1L1, found in the coating of the upper portion of cochlear hair cell stereocilia. PKHD1L1 is expressed transiently during the first two weeks or so of development, but when absent, the animal becomes progressively deaf. The protein is novel in the ear and is named for its similarity to a polycystic kidney and hepatic disease protein, but it is of unknown function and molecular identity. The authors hypothesize that this extracellular protein may be involved in linking to the many other proteins that participate in forming the external coating and various stereociliar links and they suggest the intriguing function of allowing for the gliding action of stereocilia past each other and the facilitation of bundle movement.

Many, many approaches are used to localize this novel protein and its RNA on stereocilia and to demonstrate its function in both wild type and null mice. The data figures are each outstanding, with great attention to detail, and the supplementary video is very well-done, among the best that I have seen in this area. I have no major concerns.

Specific minor comments:

P. 5, L. 20. A minor typo. I believe that the sentence should read “PKHD1L1 is a large, 4249-aa protein...”

This has been corrected.

P. 12, L. 1. I was a little confused by the statement of number of stereocilia per row. Surely there aren't 200 cilia per row? I count maybe 40 stereocilia per row in the SEMs of the P120 OHCs and fewer in the IHCs.

Thank you for spotting this confusion; it has been corrected. The data set includes a total of ~200 stereocilia of each row, not per each row (so ~600 altogether) and was collected from 6 OHCs.

P. 14, L. 15. Duly noted about the wrinkling on the TM in the nulls, but I wonder if there was any gold particle labeling for PKHD1L1 on the TM in the wild type? One supposes that this is a future direction, when examining interactions with TM proteins such as CEACAM16 and TECTA

We did do this experiment with a fluorescently labeled antibody, in a wildtype cochlea at age P7. With confocal imaging we scanned through a 24- \$\mu\$ m z stack from the top of the TM to the surface of the organ of Corti. No V-patterned fluorescence was detected in the TM, but the label was very strong at the

plane of the hair bundle of the organ of Corti. It has not been repeated enough for inclusion in this manuscript. In future studies we will repeat this with immunogold SEM at different ages.

P. 21, L. 1. Another minor typo. I suggest "...were cut as frozen sections at 10 um thickness."

This has been corrected.

References:

P. 33, L. 33-34. I suggest not using Title Case for this article title if it is not being used elsewhere.

This has been corrected.

REVIEWERS' COMMENTS:

Reviewer #1 (Remarks to the Author):

The authors have satisfactorily answered my questions and to my best judgment, the questions of the other reviewers.

Reviewer #2 (Remarks to the Author):

OK, but you should definitely check susceptibility to acoustic over exposure in the future.

Joe Santos-Sacchi

Reviewer #3 (Remarks to the Author):

The authors have satisfactorily addressed the concerns of the reviewers.